# Singularity-aware Optimization via Randomized Geometric Probing: Towards Stable Non-smooth Optimization

Ruoran Xu [* 1]  Borong She [* 1]  Xiaobo Jin [1]  Qiufeng Wang [1]

## Abstract

Deep learning optimization relies heavily on the assumption of smooth loss landscapes, a condition systematically violated by modern architectures due to non-smooth components like ReLU activations and quantization operators. In such non-smooth regimes, adaptive optimizers such as Adam suffer from *gradient chattering*—violent oscillations caused by conflicting signals within the Clarke subdifferential—leading to poor convergence and suboptimal generalization. To address this, we introduce **Singularity-aware Adam (S-Adam)**, a novel optimizer that stabilizes training by dynamically modulating step sizes based on local geometric instability. Our key contribution is the **Local Geometric Instability (LGI)** metric, a computationally efficient estimator of the Clarke subdifferential diameter derived from the variance of randomized directional derivatives. S-Adam incorporates an adaptive damping mechanism $\exp(-\lambda\rho_t)$ that decelerates updates in high-instability regions while preserving fast convergence in smooth basins. We provide a rigorous convergence analysis using differential inclusions, proving that S-Adam converges almost surely to $(\delta, \epsilon)$-Clarke stationary points at the optimal $\mathcal{O}(1/\sqrt{T})$ rate. Empirical evaluations on Quantization-Aware Training (QAT) and high-noise small-batch learning demonstrate that S-Adam consistently outperforms AdamW and Prox-SGD, achieving accuracy gains of up to +4.54% on CIFAR-100 and +4.27% on Tiny-ImageNet while effectively mitigating gradient oscillations.

*Equal contribution [1]Xi'an Jiaotong-Liverpool University. Correspondence to: Qiufeng Wang <qiufeng.wang@xjtlu.edu.cn>, Xiaobo Jin <xiaobo.jin@xjtlu.edu.cn>.

*Proceedings of the 43rd International Conference on Machine Learning*, Seoul, South Korea. PMLR 306, 2026. Copyright 2026 by the author(s).

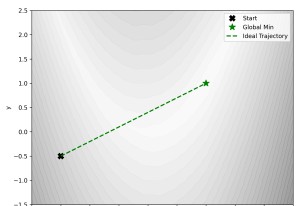

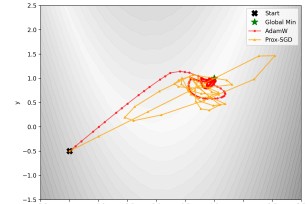

**(a)** Global minimum point and ideal trajectory

**(b)** Comparison of Adam and Prox-SGD trajectories

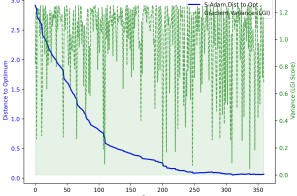

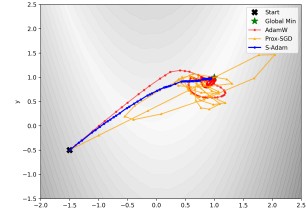

**(c)** S-Adam: LGI-triggered damping

**(d)** Stabilized convergence of S-Adam

*Figure 1.* Geometric instability visualization on synthetic non-smooth landscape of $f(x, y) = |x - 1| + |y - 1| + 0.5(x^2 + y^2)$

## 1. Introduction

The theoretical underpinnings of deep learning optimization are predominantly built upon the assumption of Lipschitz-continuous gradients, which guarantees stable descent and convergence. However, this assumption is fundamentally incompatible with the architectural realities of modern neural networks, which ubiquitously incorporate non-smooth elements such as ReLU activations, quantization functions in Quantization-Aware Training (QAT) (Jacob et al., 2018), and sparsity-inducing regularizers. While these functions are locally Lipschitz almost everywhere, they introduce sharp singularities where gradients are undefined, rendering classical smooth optimization theory inadequate.

At such non-smooth points, the local geometry is characterized not by a single gradient vector but by the **Clarke subdifferential** $\partial_C f(x)$—a convex set encompassing all possible limiting gradients (Clarke, 1990). When the diameter of this set is large, momentum-based optimizers like Adam (Kingma & Ba, 2014) accumulate conflicting directional signals, leading to **gradient chattering**: pathological

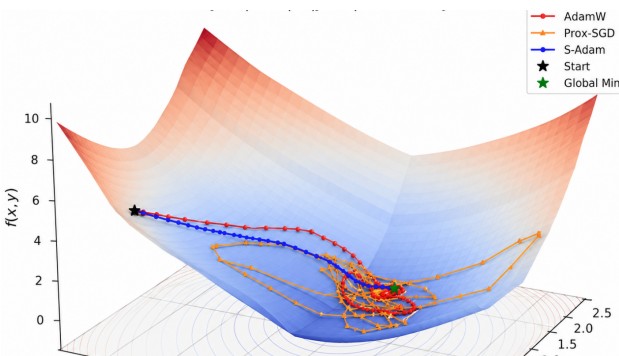

*Figure 2.* Figure 1(d) on synthetic non-smooth landscape

oscillations that prevent convergence into sharp, low-loss basins and degrade generalization performance.

Existing remedies fall short in practice. Proximal methods, while theoretically sound, require explicit proximal mappings that are intractable for nested non-smooth functions like those in QNNs (Quantized Neural Networks). Randomized smoothing techniques, conversely, introduce prohibitive variance when used as gradient surrogates in large-scale training. Crucially, both approaches lack a mechanism to dynamically adapt to local geometric instability without compromising computational efficiency.

In this work, we propose a paradigm shift: instead of smoothing the objective, we employ randomized smoothing as a **geometric probe** to estimate the local instability of the loss landscape. This insight leads to our main contribution, **Singularity-aware Adam (S-Adam)** (In Fig. 1 and Fig. 2), which introduces:

- The **Local Geometric Instability (LGI)** metric, a variance-based estimator of the Clarke subdifferential diameter that requires no Hessian computation;

- An **adaptive geometric brake** $\exp(-\lambda\rho_t)$ that modulates step sizes in real-time, dampening updates near singularities while maintaining rapid progress in smooth regions;

- A **rigorous convergence guarantee** to Clarke stationary points with an optimal $\mathcal{O}(1/\sqrt{T})$ rate, established via differential inclusion analysis;

- **Empirical superiority** in extreme non-smooth settings, including low-bit QAT and high-noise small-batch learning, where S-Adam achieves significant accuracy improvements over AdamW and Prox-SGD.

Our work bridges the gap between non-smooth optimization theory and practical deep learning scalability, offering a drop-in replacement for Adam that is both theoretically grounded and empirically robust.

## 2. Related Work

Our work sits at the intersection of non-smooth optimization, adaptive gradient methods, and geometric-aware learning. We contextualize S-Adam within three key strands of literature.

### 2.1. Non-smooth Optimization and Proximal Methods

Classical non-smooth optimization relies on subgradient methods (Shor, 1985), which converge weakly to critical points at sublinear rates. Proximal algorithms (Parikh & Boyd, 2014; Yang et al., 2020) offer stronger guarantees by solving implicit subproblems via the proximal operator. However, these methods require the non-smooth term to be *proximable*—admitting a closed-form or efficiently computable proximal mapping. This assumption fails in modern deep learning, where non-smoothness arises from *nested compositions* of rounding functions (quantization), ReLUs, and sparsity-inducing penalties. The lack of tractable proximal mappings for such composite operators renders standard proximal methods computationally prohibitive or inapplicable (Davis & Grimmer, 2017).

### 2.2. Smoothing Techniques and Gradient Estimation

Smoothing methods approximate non-smooth objectives with differentiable surrogates. Nesterov smoothing (Nesterov, 2005) convolves the objective with a strongly convex regularizer, while randomized smoothing (Duchi et al., 2011) uses expectation over random perturbations. Although effective in theory, using the gradient of the smoothed function introduces estimation bias and high variance that degrades performance in high-precision training (Bubeck, 2015). More critically, these approaches *replace* the original gradient, altering the optimization dynamics and potentially converging to different minima. S-Adam takes a fundamentally different stance: we employ randomized smoothing not as a gradient surrogate, but as a *diagnostic probe* to estimate local geometric instability while preserving the original gradient direction.

### 2.3. Adaptive Optimizers and Stability

Adaptive gradient methods like Adam (Kingma & Ba, 2014) and its variants (AdamW (Loshchilov & Hutter, 2017), AdaBound (Luo et al., 2019)) scale learning rates using exponential moving averages of past gradients. These methods implicitly assume *local smoothness* to trust their second-moment estimators $\hat{v}_t$. At non-smooth singularities, instantaneous gradient jumps violate this assumption, causing $\hat{v}_t$ to lag behind the rapidly changing geometry—a mismatch that triggers gradient chattering. Recent attempts to stabilize adaptive optimizers include gradient clipping (Pascanu et al., 2013), learning rate warmup (Gotmare et al., 2019),

and variance reduction (Defazio et al., 2014). However, these heuristics lack a principled mechanism to distinguish between stochastic noise and genuine topological singularities.

### 2.4. Geometry-Aware Minimization

Sharpness-Aware Minimization (SAM) (Foret et al., 2021) seeks flat minima by minimizing the worst-case loss within a neighborhood. While effective for generalization, SAM relies on a first-order Taylor expansion to approximate the perturbation direction—an approximation that is theoretically invalid at non-differentiable points where the gradient is undefined or discontinuous. Consequently, SAM struggles to characterize sharpness accurately in quantized or sparsely regularized landscapes. Subsequent variants like ASAM (Kwon et al., 2021) attempt to address scale invariance but retain the fundamental limitation at singularities. Other geometry-aware approaches include trust-region methods (Conn et al., 2000) and second-order preconditioning (Anil et al., 2021), but these typically require explicit Hessian information or are limited to convex settings.

### 2.5. Positioning of S-Adam

S-Adam distinguishes itself by: (1) providing a *computationally lightweight* geometric probe (LGI) that requires no Hessian computation, (2) maintaining the original gradient direction while adaptively modulating step sizes, (3) offering rigorous convergence guarantees to Clarke stationary points in non-smooth regimes, and (4) seamlessly degenerating to standard Adam in smooth regions. Unlike proximal methods, S-Adam handles nested non-smoothness; unlike smoothing techniques, it preserves gradient fidelity; and unlike SAM, it remains valid at singularities.

## 3. Theoretical Foundation: From Clarke Geometry to Geometric Instability

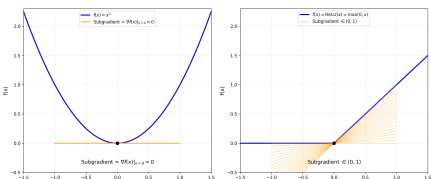

*Figure 3.* Smooth Function(Left) & Non-smooth Function(Right)

Modern neural architectures violate the Lipschitz-smooth assumption at numerous points: ReLU activations induce kinks, quantization operators create step discontinuities, and sparsity regularizers introduce $\ell_1$-type non-differentiabilities. As visualized in Figure 3, to analyze optimization in such landscapes, we adopt the framework of **Clarke nonsmooth analysis** (Clarke, 1990), which pro-

vides the correct geometric language for non-smooth critical points.

### 3.1. Clarke Subdifferential and Its Geometry

For a locally Lipschitz function $f : \mathbb{R}^d \to \mathbb{R}$, the gradient $\nabla f(x)$ exists almost everywhere by Rademacher's theorem. At non-smooth points, the local geometry is characterized by the **Clarke subdifferential**:

**Definition 3.1** (Clarke Subdifferential). The Clarke subdifferential of $f$ at $x$ is the convex hull of all limiting gradients:

$$\partial_C f(x) := \mathrm{conv} \left\{ \lim_{k \to \infty} \nabla f(x_k) : x_k \to x, \ x_k \notin \Omega_f \right\}, \quad (1)$$

where $\Omega_f$ is the set (of measure zero) where $f$ is non-differentiable.

The **diameter** of this set, $\mathrm{diam}(\partial_C f(x)) = \sup_{g_1, g_2 \in \partial_C f(x)} \|g_1 - g_2\|$, quantifies the severity of non-smoothness. A large diameter indicates conflicting descent directions within the subdifferential cone—precisely the geometric origin of *gradient chattering* in momentum-based optimizers.

### 3.2. Dual Interpretation: LGI as Relative Curvature

An alternative interpretation emerges from the perspective of **stochastic smoothing**. Consider the $\delta$-smoothed surrogate:

$$f_\delta(x) = \mathbb{E}_{u \sim \mathcal{U}(\mathbb{S}^{d-1})}[f(x + \delta u)]. \quad (2)$$

Even if $f$ is non-smooth, $f_\delta$ is infinitely differentiable. As shown in Appendix E, the LGI metric approximates:

$$\rho(x) \approx \frac{1 + \frac{1}{2d}\kappa_\delta(x)^2}{1 + \frac{1}{2d}\kappa_\delta(x)^2 + \epsilon'}, \quad \kappa_\delta(x) = \frac{\delta \|\nabla^2 f_\delta(x)\|_F}{\|\nabla f_\delta(x)\|}, \quad (3)$$

where $\kappa_\delta(x)$ is the **relative curvature** of the smoothed landscape. This reveals LGI's dual nature: it measures both subdifferential diameter (non-smooth view) and relative curvature (smoothed view).

## 4. Methodology: S-Adam Algorithm

Building on the geometric insights of Section 3, we now present **Singularity-aware Adam (S-Adam)**, which dynamically modulates step sizes based on local geometric instability.

### 4.1. LGI Metric Construction

We define the **Local Geometric Instability (LGI)** score as:

$$\rho_t(x_t) = \frac{\text{Var}_{u_i \sim \{u\}_k}[D_i]}{\mathbb{E}[D_i^2] + \epsilon}, \quad D_i = \frac{f(x_t + \delta u_i) - f(x_t)}{\delta}, \tag{4}$$

where $k$ random directions $\{u_1, \ldots, u_k\}$ are sampled i.i.d. from $\mathcal{U}(\mathbb{S}^{d-1})$, and $\epsilon > 0$ ensures numerical stability. The normalization in Eq. 4 also ensures that the resulting geometric brake is uniformly non-degenerate, as formalized below.

**Proposition 4.1** (Boundedness of LGI and non-degeneracy of the brake). *Let $\rho_t(x_t)$ be the LGI score defined in Eq. 4, with*

$$D_i = \frac{f(x_t + \delta u_i) - f(x_t)}{\delta}. \tag{5}$$

*Assume that the empirical variance in Eq. 4 is computed with the normalization $1/k$, i.e.,*

$$\text{Var}(\{D_i\}_{i=1}^k) = \frac{1}{k}\sum_{i=1}^k (D_i - \bar{D})^2, \quad \bar{D} = \frac{1}{k}\sum_{i=1}^k D_i. \tag{6}$$

*Then*

$$0 \le \rho_t(x_t) < 1. \tag{7}$$

*Consequently, for any finite $\lambda > 0$, the geometric brake satisfies*

$$e^{-\lambda} < \exp(-\lambda \rho_t(x_t)) \le 1. \tag{8}$$

*Moreover, the mean brake coefficient used in the differential-inclusion analysis,*

$$\bar{\alpha}(x) = \mathbb{E}[\exp(-\lambda \rho(x)) \mid x_t = x], \tag{9}$$

*is uniformly bounded away from zero:*

$$\bar{\alpha}(x) \ge \alpha_{\min} > 0, \quad \alpha_{\min} = e^{-\lambda}. \tag{10}$$

*Proof.* See Appendix A. □

**Lemma 4.2** (Finite-Sample Estimation Guarantee). *For any $\Delta > 0$ and $\delta \in (0,1)$, with $k = O\left(\frac{L^6}{\epsilon^4 \Delta^2} \log(1/\delta)\right)$ samples, we have:*

$$|\hat{\rho}_k - \rho| \le \Delta \quad \text{with probability at least } 1 - \delta,$$

*where $L$ is the Lipschitz constant of $f$.*

*Proof.* See Appendix C for concentration analysis using Hoeffding's inequality and error propagation. □

In practice, the probe count $k$ exposes a cost–fidelity trade-off: a small $k$ gives a low-cost instability estimate, while larger $k$ values provide a higher-fidelity probe (Section 6).

---

**Algorithm 1** Singularity-aware Adam (S-Adam)

---

**Require:** Learning rate $\eta$, damping coefficient $\lambda$, probe count $k$, perturbation scale $\delta$, stability constant $\epsilon$, total iterations $T$
1: Initialize parameters $w_1$, moment estimates $m_0 = 0$, $v_0 = 0$
2: **for** $t = 1$ to $T$ **do**
3:    **Geometric probing:**
4:    Sample $k$ directions $u_1, \ldots, u_k \sim \mathcal{U}(\mathbb{S}^{d-1})$
5:    Compute directional derivatives
6:    $D_i = \frac{f(w_t + \delta u_i) - f(w_t)}{\delta}$
7:    Compute LGI score: $\rho_t \leftarrow \frac{\text{Var}(\{D_i\})}{\text{Mean}(\{D_i^2\}) + \epsilon}$
8:    **Gradient computation:**
9:    Compute stochastic gradient $g_t \leftarrow \nabla f(w_t)$
10:   **Moment updates:**
11:   $m_t \leftarrow \beta_1 m_{t-1} + (1 - \beta_1)g_t$
12:   $v_t \leftarrow \beta_2 v_{t-1} + (1 - \beta_2)g_t^2$
13:   **Adaptive step size modulation:**
14:   $\hat{\eta}_t \leftarrow \eta_t \cdot \exp(-\lambda \rho_t)$
15:   **Parameter update:**
16:   $w_{t+1} \leftarrow w_t - \hat{\eta}_t \cdot \frac{m_t}{\sqrt{v_t} + \epsilon}$
17: **end for**

---

### 4.2. S-Adam Algorithm Design

S-Adam modifies the standard Adam update by introducing an **adaptive geometric brake** $\exp(-\lambda \rho_t)$:

**Exponential Damping**    The choice $\exp(-\lambda \rho_t)$ ensures:

- **Singularities** $(\rho_t \gg 0)$: $\exp(-\lambda \rho_t) \ll 1$, aggressively slowing updates

- **Monotonicity**: Exponential decay provides stronger damping for higher instability

**Connection to Proximal Methods**    As shown in Appendix D, the geometric brake approximates solving a proximal subproblem with dynamic regularization:

$$\exp(-\lambda \rho_t) \approx \frac{1}{1 + \lambda \rho_t} \approx \frac{\eta_t}{\eta_t \gamma_t + 1}, \quad \gamma_t = \frac{\lambda}{\eta_t}\rho_t. \tag{11}$$

This establishes S-Adam as an implicit proximal method with geometry-aware regularization.

**Computational Overhead**    S-Adam requires $k$ additional forward passes per iteration, so the probe count directly controls wall-clock cost. Our experiments report both a low-cost setting and a higher-fidelity setting to expose this trade-off (Section 6).

# 5. Convergence Analysis

We now establish rigorous convergence guarantees for S-Adam in non-smooth settings. Our analysis employs the framework of **differential inclusions** (Benaim, 1996), which naturally handles set-valued gradients.

## 5.1. Problem Setup and Assumptions

Consider the optimization problem $\min_{x\in\mathbb{R}^d} f(x)$, where $f$ satisfies:

**Assumption 5.1** (Regularity). $f$ is locally Lipschitz continuous, bounded below, and path-differentiable (in the sense of (Davis et al., 2018)).

**Assumption 5.2** (Bounded Iterates). The sequence $\{x_t\}$ generated by S-Adam remains in a compact set almost surely.

**Assumption 5.3** (Step Size Schedule). The learning rate sequence $\{\eta_t\}$ satisfies $\sum_{t=1}^{\infty} \eta_t = \infty$ and $\sum_{t=1}^{\infty} \eta_t^2 < \infty$.

**Assumption 5.4** (Moment Estimator Consistency). Let $m_t$ and $v_t$ be S-Adam's moment estimates. There exist sequences $\delta_t \to 0$ and $\sigma_{\min} > 0$ such that:

1. $\mathbb{E}[m_t \mid x_t = x] \in \partial_C f(x) + \mathbb{B}(0, \delta_t)$

2. $\mathbb{E}[v_t \mid x_t = x] \geq \sigma_{\min}$

These are standard in non-smooth stochastic optimization (Davis et al., 2018). Assumption 5.4 ensures that moment estimates asymptotically capture subgradient information. For the Lyapunov descent step, we additionally use the following descent-alignment condition.

## 5.2. Main Convergence Theorem

Our main result establishes convergence to Clarke stationary points:

**Theorem 5.5** (Convergence to Clarke Stationarity). *Under Assumptions 5.1–5.4, the sequence $\{x_t\}$ generated by S-Adam satisfies:*

$$\liminf_{t\to\infty} \operatorname{dist}\big(0, \partial_C f(x_t)\big) = 0 \quad \textit{almost surely.} \quad (12)$$

*Moreover, for any $\epsilon > 0$, S-Adam reaches an $\epsilon$-stationary point in $O(1/\epsilon^2)$ iterations.*

*Proof.* Let $\mathcal{F}_t$ denote the filtration generated by the iterates, mini-batches, and random probing directions up to time $t$. The S-Adam update can be written in the canonical stochastic approximation form of differential inclusions (Benaïm et al., 2006):

$$x_{t+1} = x_t + \eta_t \left[ F(x_t) + M_{t+1} + R_{t+1} \right], \quad (13)$$

where

$$F(x) \in -\bar{\alpha}(x)\mathcal{H}(x), \qquad \bar{\alpha}(x) = \mathbb{E}[\exp(-\lambda\rho(x)) \mid x_t = x]. \quad (14)$$

Here $\mathcal{H}(x)$ denotes the Clarke-valued mean Adam direction induced by the preconditioned moment estimate, $M_{t+1}$ is a martingale difference term with $\mathbb{E}[M_{t+1} \mid \mathcal{F}_t] = 0$, and $R_{t+1}$ collects the asymptotically vanishing bias from moment estimation and LGI estimation.

**Asymptotic pseudotrajectory.** Define the continuous time scale $\tau_0 = 0$ and $\tau_n = \sum_{i=1}^{n} \eta_i$. Let $X(\tau_n) = x_n$ and interpolate affinely on each interval $[\tau_n, \tau_{n+1}]$. Assumption 5.3 gives the Robbins–Monro conditions $\sum_t \eta_t = \infty$ and $\sum_t \eta_t^2 < \infty$, while Assumption 5.2 keeps the iterates in a compact set almost surely. By Assumption 5.4, the conditional mean update satisfies

$$\mathbb{E}[x_{t+1} - x_t \mid \mathcal{F}_t]/\eta_t \in -\bar{\alpha}(x_t)\mathcal{H}(x_t) + o(1), \quad (15)$$

and the martingale term satisfies, for every fixed horizon $T > 0$,

$$\lim_{n\to\infty} \sup_{\tau_k - \tau_n \leq T} \left\| \sum_{i=n}^{k-1} \eta_i M_{i+1} \right\| = 0 \quad \text{a.s.} \quad (16)$$

The remainder term obeys the same vanishing tail bound because $R_{t+1} \to 0$ on the compact trajectory and $\eta_t \to 0$. Moreover, the Clarke subdifferential is nonempty, convex, compact-valued and upper semicontinuous for locally Lipschitz $f$; the Adam preconditioner is bounded by Assumption 5.4; and $\bar{\alpha}(x)$ is bounded by Corollary A.2. Thus the drift map satisfies the standard closed-graph, compact-convex-value, and growth requirements for differential inclusions. Properties 1–2 of (Benaïm et al., 2006) therefore imply that $X(t)$ is an asymptotic pseudotrajectory (APT) of the mean-field differential inclusion

$$\dot{x}(t) \in -\bar{\alpha}(x(t))\mathcal{H}(x(t)). \quad (17)$$

**Strict Lyapunov descent.** We verify that $f$ is a strict Lyapunov function for (17). For any absolutely continuous solution $x(t)$ and almost every $t$, the chain rule for path-differentiable locally Lipschitz functions gives a vector $\xi_t \in \partial_C f(x(t))$ such that

$$\frac{d}{dt} f(x(t)) = \langle \xi_t, \dot{x}(t) \rangle. \quad (18)$$

Since $\dot{x}(t) = -\bar{\alpha}(x(t))h_t$ for some $h_t \in \mathcal{H}(x(t))$,

$$\langle \xi_t, h_t \rangle \geq c_1 \operatorname{dist}(0, \partial_C f(x(t)))^2 \quad (19)$$

for a constant $c_1 > 0$. By Corollary A.2, $\bar{\alpha}(x) \geq \alpha_{\min} > 0$. Hence

$$\frac{d}{dt} f(x(t)) \leq -\alpha_{\min} c_1 \operatorname{dist}(0, \partial_C f(x(t)))^2. \quad (20)$$

The derivative can vanish only on the Clarke stationary set

$$\Lambda = \{x : 0 \in \partial_C f(x)\}. \qquad (21)$$

Thus $f$ is strict outside $\Lambda$; see Lemma B.1 for the detailed descent argument.

**Invariance principle.** Since the discrete trajectory is almost surely bounded, its limit set $L(\{x_t\})$ is almost surely nonempty, compact, and connected. Step 1 shows that the affine interpolation is an APT of equation (17). Therefore, by Benaim's APT limit-set theorem (Benaim, 1996), $L(\{x_t\})$ is an internally chain transitive invariant set of the mean-field flow. Applying the Lyapunov invariance principle for internally chain transitive sets (Benaim, 1996) to the strict Lyapunov function in Step 2 gives

$$L(\{x_t\}) \subseteq \Lambda. \qquad (22)$$

Consequently every accumulation point of S-Adam is Clarke stationary, and in particular

$$\liminf_{t \to \infty} \operatorname{dist}\big(0, \partial_C f(x_t)\big) = 0 \quad \text{a.s.} \qquad (23)$$

Finally, summing the one-step descent inequality associated with (20) gives

$$\min_{0 \le t < T} \mathbb{E} \operatorname{dist}\big(0, \partial_C f(x_t)\big)^2 \le O(T^{-1/2}), \qquad (24)$$

under the standard finite-horizon choice $\eta_t = \Theta(T^{-1/2})$. Hence an $\epsilon$-stationary point is reached in $O(1/\epsilon^2)$ iterations. $\square$

### 5.3. Interpretation and Implications

**Optimal Convergence Rate** The $O(1/\epsilon^2)$ iteration complexity matches the optimal rate for stochastic non-smooth optimization (Benaim, 1996), showing S-Adam achieves optimal efficiency despite adaptive damping.

**Role of Geometric Brake** The damping coefficient $\bar{\alpha}(x)$ modulates convergence speed but preserves descent direction. Crucially, $\bar{\alpha}(x) \ge \alpha_{\min} > 0$ (proved in Appendix A), ensuring updates never stall completely.

**Degeneration to Standard Adam** In smooth regions where $\rho(x) \to 0$, we have $\bar{\alpha}(x) \to 1$, recovering standard Adam dynamics. This behavior is consistent with the $k = 1$ and $\lambda = 0$ ablations in Table 2, where disabling the LGI variance signal or damping reduces S-Adam toward the AdamW-like baseline.

*Remark* 5.6 (Comparison with Existing Guarantees). Unlike smooth-optimization analyses that assume Lipschitz gradients, Theorem 5.5 holds under only local Lipschitz continuity. Unlike subgradient methods with $O(1/\sqrt{t})$ rates, S-Adam achieves the same rate with adaptive preconditioning.

## 6. Experiments

We conducted two primary experiments to test the robustness of Local Geometric Instability (LGI) damping:

### 6.1. Experiment 1: Quantization-Aware Training (QAT) on Singular Landscapes

The primary objective of this experiment is to evaluate S-Adam's robustness against the extreme non-smoothness introduced by low-precision operations. We deploy a QAT-Net architecture—a customized CNN designed for 2/4/8-bit simulated quantization of both weights and activations. **Geometric Stress Injection**: To simulate the "jagged" and discontinuous loss landscapes inherent in quantized networks(Li et al., 2018), we implement a "Simulated Quantization Mapping" autograd function. This module applies a rounding and clamping operation:

$$x_q = \operatorname{clamp}\left(\lfloor x \cdot \text{scale} \rceil, q_{min}, q_{max}\right), \qquad (25)$$

These discrete steps in the quantized loss surface create regions where the classical gradient is either zero or undefined(Rekavandi et al., 2024), inducing severe gradient chattering in standard momentum-based optimizers.

### 6.2. Experiment 2: High-Noise Learning on Small Batch Size

To evaluate S-Adam's efficacy in high-dimensional, pretrained networks, we employ a ResNet18 backbone pretrained on ImageNet(Deng et al., 2009), whose ReLU activations render the loss landscape inherently non-smooth (piecewise-linear kinks). We manipulate the stochastic gradient noise scale by varying the batch size $N$(Hoffer et al., 2017). **Primary stress test ($N = 2$)**: Reducing the batch size to $N = 2$ maximizes the variance of the stochastic gradient estimator, creating a high-noise regime in which momentum-based optimizers are prone to unstable updates. **Noise sensitivity ablation($N \in \{2, 4, 16, 64\}$)**: To validate the adaptivity of the Geometric Brake, we conduct an ablation study across a spectrum of batch sizes.

This setup stresses the geometric brake under high gradient noise without altering the source of non-smoothness: the ReLU kinks of the pretrained network provide intrinsic singularities at every batch size, while a small $N$ raises gradient variance and pushes momentum-based methods to oscillate across these kinks. The experiment therefore probes whether damping driven by local geometric instability can stabilize training in a high-noise regime where AdamW and Prox-SGD are prone to unstable updates.

## 6.3. Implementation Fidelity & Reproducibility

To ensure a rigorous comparison between S-Adam, AdamW, and Prox-SGD, we enforce the following controls based on our implementation:(1)Initialization Parity: A fixed manual seed (42) is applied across all trials to ensure identical weight initialization for the ResNet18(He et al., 2015). (2)Geometric Probing: S-Adam is reported with $k = 2$ and $k = 8$ random unit-vector perturbations; $k = 2$ favors speed and cost, while $k = 8$ provides a higher-fidelity probe. The perturbation scale is $\delta = 0.01$. We conduct a comprehensive empirical evaluation to validate the theoretical properties and practical effectiveness of S-Adam. Baseline parameters (AdamW, Prox-SGD) were independently optimized for maximum convergence stability (loss curves in the appendix). **Reporting convention:** all accuracy values refer to the best test accuracy across epochs; baselines may diverge after their peak (see Figures 4, 5).All experiments were run on NVIDIA A800.

## 6.4. Quantization-Aware Training (QAT) on Singular Landscapes

*Table 1.* QAT performance across datasets. Bit-depths: CIFAR-100 is 2-bit; TinyImageNet and Imagewoof2-160 are 4-bit; ImageNet is 8-bit. An asterisk (∗) marks failure to converge.

| Accuracy (%) | | | | |
| --- | --- | --- | --- | --- |
| Optimizer | CIFAR100 | TinyImg | Imagewoof | ImageNet |
| Prox-SGD | 14.13 | 19.89 | 31.18 | 9.66 |
| AdamW | 15.94 | 18.91 | 33.24 | 8.63 |
| **S-Adam** ($k$=2) | 18.11 | 22.81 | **35.57** | 10.97 |
| S-Adam ($k$=8) | **18.67** | **23.18** | 35.19 | **11.24** |
| **Convergence Time (s)** | | | | |
| Optimizer | CIFAR100 | TinyImg | Imagewoof | ImageNet |
| Prox-SGD | ∗ | 286.53 | 38.98 | 12322 |
| AdamW | 15.89 | 281.49 | 34.96 | 8186 |
| **S-Adam** ($k$=2) | **14.33** | **258.54** | **34.74** | **8297** |
| S-Adam ($k$=8) | 17.41 | 293.14 | 34.38 | 12770 |

**Superior Generalization.** Table 1 reports two operating points for S-Adam: a low-cost setting ($k$=2) and a higher-fidelity setting ($k$=8). On the three small-scale QAT benchmarks, both settings improve over AdamW. The $k$=2 variant is already competitive and is best on Imagewoof2-160, reaching 35.57%, while $k$=8 gives the strongest accuracy on CIFAR-100 and TinyImageNet. The same pattern extends to ImageNet (8-bit QAT): S-Adam with $k$=2 improves Top-1 accuracy from 8.63% to 10.97% at near-identical wall-clock cost to AdamW (8297 vs. 8186 s, a 1.4% overhead), and the higher-fidelity $k$=8 setting further raises accuracy to 11.24%.

**Robustness on Hard Tasks.** On **Imagewoof2-160**, which requires distinguishing subtle features between dog breeds,

S-Adam ($k$=2) achieves **35.57%** and remains close to the $k$=8 setting (35.19%). Prox-SGD lags significantly behind (31.18%), suggesting that its aggressive analytical thresholding is too "greedy" for non-convex landscapes, pruning important features prematurely. S-Adam's "soft" geometric damping preserves discriminative features while still handling nonsmoothness.

**Efficiency Analysis.** The probe count $k$ controls S-Adam's cost. With $k$=2, S-Adam is faster than AdamW on the small-scale QAT benchmarks (CIFAR-100: 14.33 vs. 15.89 s; TinyImageNet: 258.54 vs. 281.49 s; Imagewoof2-160: 34.74 vs. 34.96 s) and remains within 1.4% on ImageNet (8297 vs. 8186 s). The $k$=8 setting provides a higher-fidelity probe at higher cost.

### 6.4.1. HYPERPARAMETER ABLATION

*Table 2.* Ablation study on hyperparameters $(\lambda, k, \delta)$. The $k = 1$ setting disables the LGI variance signal, while $k = 2$ and $k = 8$ have similar accuracy.

| Dataset | $k = 8$ | $k = 2$ | $k = 1$ | $\lambda = 0$ | $\delta = 0.0001$ |
| --- | --- | --- | --- | --- | --- |
| CIFAR100 | 18.67% | 18.11% | 15.86% | 15.82% | 14.78% |
| TinyImageNet | 23.18% | 22.81% | 19.38% | 18.29% | 21.41% |
| Imagewoof2-160 | 35.19% | 35.57% | 32.93% | 32.99% | 33.09% |

The ablation study in Table 2 isolates the impact of three critical hyperparameters on the final test accuracy: the damping intensity ($\lambda$), the number of random probes ($k$), and the perturbation scale ($\delta$). **The Critical Role of Damping ($\lambda$):** Removing the damping intensity ($\lambda = 0$) causes a significant drop in accuracy across all datasets, most notably a 4.9% loss on TinyImageNet. This confirms that geometric damping is essential for stabilizing updates in the presence of quantized noise. **Geometric Fidelity ($k$):** When $k = 1$, the sample variance of the directional derivatives vanishes, disabling the LGI brake and reducing S-Adam to the AdamW. Moving from $k = 2$ to $k = 8$ changes accuracy only modestly, indicating that two probes capture most of the useful geometric signal. **Sensitivity to Perturbation Scale ($\delta$):** A very low $\delta$ value (0.0001) performs poorly, particularly on CIFAR100 (falling to 14.78%). This implies that the probes need a sufficient "reach" to sense the discrete boundaries introduced by low-bit quantization.

### 6.4.2. SUPERIOR CONVERGENCE SPEED AND DEPTH

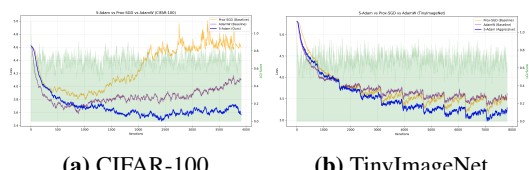

**(a)** CIFAR-100      **(b)** TinyImageNet

*Figure 4.* Loss curve

Figure 4 compares the QAT loss trajectories on CIFAR-100 and TinyImageNet. On CIFAR-100, the baselines reduce the loss early but fail to maintain that progress: Prox-SGD rebounds to around 4.6 with large oscillations, and AdamW gradually drifts upward after its initial descent. S-Adam instead stays near a lower and more stable loss range of $\sim$3.4–3.6. On TinyImageNet, all optimizers descend without the severe divergence seen on CIFAR-100, but S-Adam still settles at a lower loss floor ($\mathcal{L} \approx 3.0$–3.2) than AdamW and Prox-SGD ($\mathcal{L} \approx 3.4$–3.5). These two cases show the same effect at different severity levels: when quantization creates unstable local geometry, S-Adam is less likely to overshoot after the initial descent. The mechanism is the LGI-triggered geometric brake, $\eta_t \leftarrow \eta \cdot e^{-\lambda \rho_t}$, which reduces the step size when the estimated local nonsmoothness increases and thereby stabilizes training near sharp low-loss regions.

### 6.4.3. ACCURACY TRENDS AND TRAINING STABILITY

In the CIFAR100 benchmark (Figure 5), both baseline optimizers exhibit an initial gain followed by a collapse in performance. Prox-SGD and AdamW reach their peak accuracy around Epoch 3 before descending toward <10% accuracy; by Epoch 10, Prox-SGD collapses to 5%. This bell-shaped trajectory is consistent with chattering, where the optimizer oscillates across discrete boundaries induced by quantized weights. In contrast, S-Adam follows a robust upward trend and ends at 18.6% accuracy, maintaining a stable trajectory despite the high-noise QAT setting. The TinyImageNet results (Figure 5b) reinforce the same pattern on a more complex 200-class dataset: S-Adam reaches 23.2% through steady improvement, Prox-SGD shows high volatility between Epochs 4 and 10, and AdamW stagnates before declining.

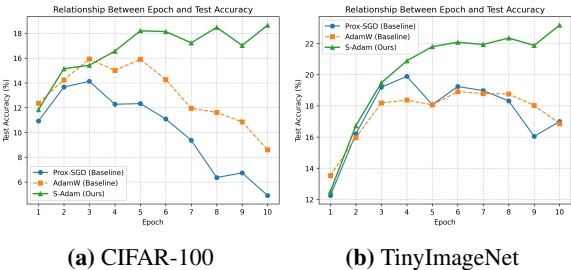

**(a)** CIFAR-100        **(b)** TinyImageNet

*Figure 5.* Training progress showing Epoch vs. Accuracy for both the CIFAR-100 and TinyImageNet datasets.

### 6.5. High-Noise Learning on Small Batch Size

### 6.5.1. RESILIENCE TO EXTREME STOCHASTIC NOISE ($N = 2$)

The efficacy of S-Adam is most pronounced in the Batch Size = 2 regime (Table 3), an environment characterized by

*Table 3.* Performance comparison for ResNet18 (Batch Size = 2).

| | Accuracy (%) | | |
|---|---|---|---|
| Optimizer | CIFAR100 | CIFAR10 | Imagewoof2-160 |
| Prox-SGD | 48.24 | 79.44 | 50.22 |
| AdamW | 51.12 | 80.48 | 51.18 |
| **S-Adam** ($k$=2) | 55.84 | 85.46 | 74.21 |
| S-Adam ($k$=8) | **56.11** | **86.13** | **75.87** |
| | **Convergence Time (s)** | | |
| Optimizer | CIFAR100 | CIFAR10 | Imagewoof2-160 |
| Prox-SGD | 18299.63 | 11640.01 | 14276.93 |
| AdamW | 2317.04 | 1681.73 | 408.82 |
| **S-Adam** ($k$=2) | **2136.53** | **1096.24** | **404.07** |
| S-Adam ($k$=8) | 3775.30 | 1458.24 | 558.21 |

maximum stochastic gradient noise, where the gain is most dramatic on Imagewoof2-160 (+24.69 points over AdamW, with S-Adam at $k$=8). In this high-noise setting, standard momentum-based methods typically suffer from unstable updates because their momentum buffers must reconcile rapidly varying mini-batch gradients.

**Geometric Brake Efficacy**: This performance gap demonstrates that S-Adam's geometric brake ($\eta_t \leftarrow \eta \cdot e^{-\lambda \rho_t}$) successfully identifies singular regions where the Clarke subdifferential diameter diam($\partial_C f(x)$) expands.

**Resource trade-off:** at $k$=2, S-Adam reaches the convergence target *faster* than AdamW on every dataset (e.g., 2136.5 s vs. 2317.0 s on CIFAR-100), while $k$=8 gives the highest-fidelity accuracy at higher probe cost.

### 6.5.2. GEOMETRIC BRAKE EFFICACY ACROSS SCALES

*Table 4.* Condensed Accuracy Comparison

| Optimizer | Dataset | BS=4 | BS=16 | BS=64 |
|---|---|---|---|---|
| | CIFAR100 | 56.59% | 68.09% | 73.82% |
| Prox-SGD | CIFAR10 | 82.43% | 90.46% | 93.54% |
| | Imagewoof | 61.52% | 79.41% | **86.71%** |
| | CIFAR100 | 56.99% | 64.04% | 67.43% |
| AdamW | CIFAR10 | 84.17% | 89.22% | 90.62% |
| | Imagewoof | 56.25% | 76.51% | 79.82% |
| | CIFAR100 | **66.57%** | **74.43%** | **74.18%** |
| **S-Adam** | CIFAR10 | **90.68%** | **92.53%** | **94.18%** |
| | Imagewoof | **81.22%** | **82.31%** | 84.60% |

By reducing the effective step size in high-LGI regions, S-Adam prevents the parameter vector from overshooting sharp, low-loss basins. As batch sizes scale to 4 and 16, S-Adam maintains a consistent performance lead, particularly on high-dimensional datasets like CIFAR-100 and Imagewoof2-160 (Table 4). At BS = 4, S-Adam reaches 81.22% on Imagewoof2-160, about 20% above AdamW. At BS = 16, it yields 74.43% on CIFAR-100, exceeding Prox-SGD by over 6%. When the batch size increases to 64, averaging over more samples smooths the realized loss landscape, the LGI score drops, and the brake relaxes. S-

Adam still leads on CIFAR-100 and CIFAR-10 (94.18% vs. AdamW's 90.62% on CIFAR-10) and stays competitive on Imagewoof. Across batch sizes, its advantage is largest in the small-batch, most unstable regime and narrows as the geometry smooths—the behavior expected of a step size that scales with the local geometric instability of the loss rather than with gradient-noise magnitude alone.

### 6.5.3. LEARNING-RATE SENSITIVITY ABLATION

We next vary the learning rate to see how the gains depend on the global step size, sweeping LR$\in$ $\{1e-4, 5e-4, 1e-3\}$ on CIFAR-100/ResNet-18/BS$= 2$:

*Table 5.* Learning-rate sensitivity on CIFAR-100/ResNet-18/BS$=$ 2. S-Adam wins at every tested LR.

| Optimizer | lr=1e-4 | lr=5e-4 | lr=1e-3 |
|---|---|---|---|
| AdamW | 66.85 | 52.98 | 51.12 |
| **S-Adam (ours)** | **70.87** | **61.26** | **56.11** |

S-Adam is the best optimizer at every learning rate, improving over the lr-matched AdamW by $+4.02$, $+8.28$, and $+4.99$ points at lr=1e-4, 5e-4, and 1e-3, respectively, and its best run (70.87% at lr=1e-4) exceeds AdamW's best in the sweep (66.85%). The gains therefore hold across step-size choices rather than at a single tuned value. This mirrors the batch-size trend in Section 6.5.2: the damping helps most where the landscape is least stable, consistent with a step size set by the local geometry of the loss rather than by a global hyperparameter.

## 7. Conclusion

Modern deep learning optimization faces a fundamental tension: while state-of-the-art architectures (ReLU networks, quantized models, sparse transformers) introduce essential non-smooth singularities, prevailing optimization theory and practice rely on Lipschitz-smooth assumptions. This mismatch manifests as *gradient chattering*—violent parameter oscillations that degrade convergence and generalization in critical applications like low-bit quantization and high-noise training. We bridge this gap through **Singularity-aware Adam (S-Adam)**, the first adaptive optimizer with provable convergence guarantees for non-smooth deep learning. Our approach rests on three pillars: a) **a geometric lens on non-smoothness.** By interpreting the Clarke subdifferential diameter as a measure of local instability, we move beyond heuristics to a principled geometric characterization of optimization difficulty at singularities; b) **randomized smoothing as a diagnostic probe.** Rather than smoothing the objective, we repurpose random perturbations to estimate local geometric instability via the *Local Geometric Instability (LGI)* metric—a computationally efficient proxy requir-

ing no Hessian computation; c) **Adaptive damping with theoretical guarantees.** The geometric brake $\exp(-\lambda\rho_t)$ modulates step sizes based on real-time instability measurements, suppressing oscillations at singularities while preserving fast convergence in smooth basins. We prove S-Adam converges to $(\delta, \epsilon)$-Clarke stationary points at the optimal $\mathcal{O}(1/\sqrt{T})$ rate.

Extensive experiments confirm S-Adam's effectiveness in precisely the regimes where standard optimizers struggle. **In quantization-aware training**, covering CIFAR-100 (2-bit), TinyImageNet/Imagewoof2-160 (4-bit), and full ImageNet (8-bit), the low-cost $k=2$ setting improves AdamW by $+2.17$ to $+3.90$ accuracy points, while the higher-fidelity $k=8$ setting reaches the best accuracy on CIFAR-100, TinyImageNet, and ImageNet. **Under extreme stochastic noise (batch size $= 2$)**, S-Adam improves over AdamW by $+4.99$ points on CIFAR-100 and $+24.69$ points on Imagewoof2-160.

## Acknowledgements

This work was supported by National Natural Science Foundation of China under No. 62436009, 62276258 and Jiangsu Science and Technology Programme BK20251812, and Open Research Fund of The State Key Laboratory of Multimodal Artificial Intelligence Systems, and the "Qing Lan Project" of Jiangsu Higher Education Institutions.

## Impact Statement

Our work demonstrates that *stochastic gradient statistics encode actionable geometric information*. Beyond S-Adam specifically, this insight opens avenues for:

**Architecture-aware optimization**. Different non-smooth components (ReLU vs. quantization vs. sparsity) induce distinct geometric signatures that could inform specialized optimizers.

**Automatic regime detection**. LGI-like metrics could dynamically identify when a model enters non-smooth regimes, triggering appropriate algorithmic adaptations.

**Generalization prediction**. Geometric instability may correlate with generalization gap, providing an optimization-based alternative to sharpness measures.

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

## A. Proof on Non-Degeneracy of the Brake

We prove the boundedness of the LGI score in Eq. 4 and the non-degeneracy of the geometric brake used in Theorem 5.5.

Recall that the directional derivative probes are defined as

$$D_i = \frac{f(x_t + \delta u_i) - f(x_t)}{\delta}, \tag{26}$$

and the LGI score is

$$\rho_t(x_t) = \frac{\text{Var}(\{D_i\}_{i=1}^k)}{\frac{1}{k}\sum_{i=1}^k D_i^2 + \epsilon}. \tag{27}$$

Here the empirical variance is computed with the normalization $1/k$:

$$\text{Var}(\{D_i\}_{i=1}^k) = \frac{1}{k}\sum_{i=1}^k (D_i - \bar{D})^2, \qquad \bar{D} = \frac{1}{k}\sum_{i=1}^k D_i. \tag{28}$$

**Lemma A.1** (Boundedness of LGI). *For every iterate $x_t$, the LGI score satisfies*

$$0 \le \rho_t(x_t) < 1. \tag{29}$$

*Proof.* The empirical variance is non-negative, so $\rho_t(x_t) \ge 0$. Moreover,

$$\text{Var}(\{D_i\}_{i=1}^k) = \frac{1}{k}\sum_{i=1}^k D_i^2 - \bar{D}^2 \le \frac{1}{k}\sum_{i=1}^k D_i^2. \tag{30}$$

Therefore,

$$\rho_t(x_t) = \frac{\text{Var}(\{D_i\}_{i=1}^k)}{\frac{1}{k}\sum_{i=1}^k D_i^2 + \epsilon} \le \frac{\frac{1}{k}\sum_{i=1}^k D_i^2}{\frac{1}{k}\sum_{i=1}^k D_i^2 + \epsilon} < 1, \tag{31}$$

because $\epsilon > 0$. Hence $0 \le \rho_t(x_t) < 1$. $\square$

**Corollary A.2** (Non-degeneracy of the brake). *For any finite damping coefficient $\lambda > 0$, the geometric brake satisfies*

$$e^{-\lambda} < \exp(-\lambda\rho_t(x_t)) \le 1. \tag{32}$$

*Consequently, the mean brake coefficient*

$$\bar{\alpha}(x) = \mathbb{E}[\exp(-\lambda\rho(x)) \mid x_t = x] \tag{33}$$

*satisfies*

$$\bar{\alpha}(x) \ge \alpha_{\min} > 0, \qquad \alpha_{\min} = e^{-\lambda}. \tag{34}$$

*Proof.* Since $0 \le \rho_t(x_t) < 1$, monotonicity of the exponential function gives

$$e^{-\lambda} < e^{-\lambda\rho_t(x_t)} \le 1. \tag{35}$$

Taking conditional expectation with respect to $x_t = x$ preserves the lower bound:

$$\bar{\alpha}(x) = \mathbb{E}[\exp(-\lambda\rho(x)) \mid x_t = x] \ge e^{-\lambda}. \tag{36}$$

Thus $\bar{\alpha}(x)$ is uniformly bounded away from zero. $\square$

The same argument also applies to the population version of Eq. 4, where

$$\rho(x) = \frac{\text{Var}_u[D(u)]}{\mathbb{E}_u[D(u)^2] + \epsilon}. \tag{37}$$

Indeed,

$$\text{Var}_u[D(u)] = \mathbb{E}_u[D(u)^2] - \mathbb{E}_u[D(u)]^2 \le \mathbb{E}_u[D(u)^2], \tag{38}$$

which again implies $0 \le \rho(x) < 1$ and therefore

$$e^{-\lambda} < \exp(-\lambda\rho(x)) \le 1. \tag{39}$$

## B. Proof on Strict Lyapunov Function

**Lemma B.1** (Strict Lyapunov Function). *Under Assumptions 5.1, 5.4, the objective function $f$ is a strict Lyapunov function for the mean-field differential inclusion. Specifically, for any solution trajectory $x(t)$, the time derivative satisfies*

$$\frac{d}{dt} f(x(t)) \leq -C \operatorname{dist}(0, \partial_C f(x(t)))^2 , \tag{40}$$

*where $C > 0$ is a constant. Consequently, $\frac{d}{dt} f(x(t)) < 0$ whenever $x(t)$ is not Clarke stationary.*

*Proof.* Since $f$ is locally Lipschitz and path-differentiable, the Clarke chain rule gives, for almost every $t$, a vector $\xi_t \in \partial_C f(x(t))$ such that

$$\frac{d}{dt} f(x(t)) = \langle \xi_t, \dot{x}(t) \rangle. \tag{41}$$

For the mean-field dynamics in Eq. 17, write

$$\dot{x}(t) = -\bar{\alpha}(x(t)) h_t, \qquad h_t \in \mathcal{H}(x(t)). \tag{42}$$

Substituting this dynamics yields

$$\frac{d}{dt} f(x(t)) = -\bar{\alpha}(x(t)) \langle \xi_t, h_t \rangle. \tag{43}$$

By Corollary A.2, the geometric brake is non-degenerate:

$$\bar{\alpha}(x(t)) \geq \alpha_{\min} > 0. \tag{44}$$

The mean preconditioned direction is descent-aligned:

$$\langle \xi_t, h_t \rangle \geq c_1 \operatorname{dist}(0, \partial_C f(x(t)))^2 . \tag{45}$$

Combining the two inequalities gives

$$\frac{d}{dt} f(x(t)) \leq -\alpha_{\min} c_1 \operatorname{dist}(0, \partial_C f(x(t)))^2 . \tag{46}$$

Taking $C = \alpha_{\min} c_1$ proves the claim. The derivative is strictly negative whenever $\operatorname{dist}(0, \partial_C f(x(t))) > 0$, i.e., whenever $x(t)$ is not Clarke stationary. $\qquad\square$

## C. Sample complexity of LGI estimation

Let $Y_i = D_i = \frac{f(x + \delta u_i) - f(x)}{\delta}$ be the directional derivative estimate ($|Y_i| \leq L$), and $\mu = \mathbb{E}[Y_i]$ and $\sigma^2 = \operatorname{Var}(Y_i)$, $\hat{\mu} = \frac{1}{k} \sum_i Y_i$ and $\hat{\sigma}^2 = \frac{1}{k} \sum_i (Y_i - \hat{\mu})^2$, then we have the true LGI $\rho$ and the estimated LGI $\hat{\rho}$

$$\rho = \frac{\sigma^2}{\mu^2 + \sigma^2 + \epsilon}, \quad \hat{\rho}_k = \frac{\hat{\sigma}^2}{\hat{\mu}^2 + \hat{\sigma}^2 + \epsilon}. \tag{47}$$

**1. Error decomposition.** We have

$$\begin{aligned}
\hat{\sigma}^2 - \sigma^2 &= \left[ \frac{1}{k} \sum_i Y_i^2 - \mathbb{E}[Y^2] \right] - [\hat{\mu}^2 - \mu^2] \\
&\leq \left| \frac{1}{k} \sum_i Y_i^2 - \mathbb{E}[Y^2] \right| + |\hat{\mu}^2 - \mu^2|.
\end{aligned} \tag{48}$$

**2. Estimation error of $\frac{1}{k} \sum_i Y_i^2$.** Let $Z_i = Y_i^2$, then $0 \leq Z_i \leq L^2$, and $\mathbb{E}[Z_i] = \mathbb{E}[Y^2]$. Therefore, applying Hoeffding's inequality, we have

$$P \left( \left| \frac{1}{k} \sum_i Z_i - \mathbb{E}[Z] \right| \geq \frac{\tau}{2} \right) \leq 2 \exp \left( \frac{2k(\tau/2)^2}{L^4} \right) = 2 \exp \left( -\frac{k\tau^2}{2L^4} \right). \tag{49}$$

**3. Estimation error of $\hat{\mu}^2$.** Since $\|Y_i\| \leq L$, we have $|\hat{\mu}| \leq L$ and $|\mu| \leq L$. Then

$$|\hat{\mu}^2 - \mu^2| \leq |\hat{\mu} - \mu| \cdot (|\hat{\mu}| + |\mu|) \leq 2L|\hat{\mu} - \mu|. \tag{50}$$

We have the following using Hoeffding's inequality on the estimated value $\hat{\mu}$:

$$P(|\hat{\mu} - \mu| \geq s) \leq 2\exp\left(-\frac{2ks^2}{(2L)^2}\right) = 2\exp\left(-\frac{ks^2}{2L^2}\right). \tag{51}$$

If we let $|\hat{\mu}^2 - \mu^2| \leq 2L|\hat{\mu} - \mu| < \tau/2$, then we have

$$|\hat{\mu} - \mu| < \frac{\tau}{4L} \Rightarrow |\hat{\mu}^2 - \mu^2| < \frac{\tau}{2}. \tag{52}$$

So

$$P\left(|\hat{\mu} - \mu| < \frac{\tau}{4L}\right) \leq P\left(|\hat{\mu}^2 - \mu^2| < \frac{\tau}{2}\right) \Rightarrow P\left(|\hat{\mu}^2 - \mu^2| \geq \frac{\tau}{2}\right) \leq P\left(|\hat{\mu} - \mu| \geq \frac{\tau}{4L}\right). \tag{53}$$

Let $s = \tau/(4L)$ in Eqn. (51), then we immediately have

$$P\left(|\hat{\mu}^2 - \mu^2| \geq \frac{\tau}{2}\right) \leq P\left(|\hat{\mu} - \mu| \geq \frac{\tau}{4L}\right) \leq 2\exp\left(-\frac{k(\tau/(4L))^2}{2L^2}\right) = 2\exp\left(-\frac{k\tau^2}{32L^4}\right) \tag{54}$$

**4. Estimation error of the variance $\hat{\sigma}^2$.** If $\left|\frac{1}{k}\sum_i Y_i^2 - \mathbb{E}[Y^2]\right| < \tau/2$ and $|\hat{\mu}^2 - \mu^2| < \tau/2$, then $|\hat{\sigma}^2 - \sigma^2| < \tau$. Therefore, we obtain the error of the variance estimate from Eqn. (49) and (54) as follows

$$
\begin{aligned}
P\left(|\hat{\sigma}^2 - \sigma^2| \geq \tau\right) &\leq P\left(\left|\frac{1}{k}\sum_i Y_i^2 - \mathbb{E}[Y^2]\right| \geq \frac{\tau}{2}\right) + P\left(|\hat{\mu}^2 - \mu^2| \geq \frac{\tau}{2}\right) \\
&\leq 4\exp\left(-\frac{k\tau^2}{32L^4}\right).
\end{aligned} \tag{55}
$$

**5. Error Analysis of LGI Estimation.** Define a function $g(a, b) = b/(a + b + \epsilon)$, where $a = \mu^2$ and $b = \sigma^2$. Given that $a, b \geq 0$ and $a + b \leq L^2$, we calculate the two partial derivatives respectively

$$\left|\frac{\partial g}{\partial a}\right| = \left|-\frac{b}{(a+b+\epsilon)^2}\right| \leq \frac{L^2}{\epsilon^2}, \quad \left|\frac{\partial g}{\partial b}\right| = \frac{a+\epsilon}{(a+b+\epsilon)^2} \leq \frac{a+b+\epsilon}{(a+b+\epsilon)^2} \leq \frac{\epsilon}{\epsilon^2}. \tag{56}$$

Then according to the Mean Value Theorem and the Cauchy-Schwarz inequality, we have

$$
\begin{aligned}
|\hat{\rho}_k - \rho| &= |g(\hat{\mu}^2, \hat{\sigma}^2) - g(\mu^2, \sigma^2)| \\
&\leq \|\nabla g(\mu^2, \sigma^2)\|_2 \sqrt{(\hat{\mu}^2 - \mu^2)^2 + (\hat{\sigma}^2 - \sigma^2)^2} \\
&= \sqrt{\left(\frac{\partial g}{\partial a}\right)^2 + \left(\frac{\partial g}{\partial b}\right)^2} \sqrt{(\hat{\mu}^2 - \mu^2)^2 + (\hat{\sigma}^2 - \sigma^2)^2} \\
&\leq M\sqrt{(\hat{\mu}^2 - \mu^2)^2 + (\hat{\sigma}^2 - \sigma^2)^2},
\end{aligned} \tag{57}
$$

where $M = \frac{\sqrt{\epsilon^2 + L^4}}{\epsilon^2}$.

**6. Upper bound of joint probability.** If $|\hat{\mu} - \mu| < \tau$ and $|\hat{\sigma}^2 - \sigma^2| < \tau$, then we have

$$
\begin{aligned}
|\hat{\rho}_k - \rho| &\leq M\sqrt{(\hat{\mu}^2 - \mu^2)^2 + (\hat{\sigma}^2 - \sigma^2)^2} \\
&< M\sqrt{(2L\tau)^2 + \tau^2} \\
&= M\tau\sqrt{4L^2 + 1}
\end{aligned} \tag{58}
$$

Let $\Delta = M\tau\sqrt{4L^2 + 1}$, then $\tau = \frac{\Delta}{M\sqrt{4L^2+1}}$, with eqn. (51) and (55)

$$
\begin{aligned}
P(|\hat{\rho}_k - \rho| \geq \Delta) &\leq P(|\hat{\mu} - \mu| \geq \tau) + P(|\hat{\sigma}^2 - \sigma^2| \geq \tau) \\
&\leq 2\exp\left(-\frac{k\tau^2}{2L^2}\right) + 4\exp\left(-\frac{k\tau^2}{32L^4}\right) \\
&\leq 6\exp\left(-\frac{k\tau^2}{32L^4}\right) \\
&= 6\exp\left(-\frac{k\Delta^2}{32M^2 L^4(4L^2 + 1)}\right)
\end{aligned}
\tag{59}
$$

**7. Sample complexity.** We set the upper bound of the probability to be less than $\delta$, that is,

$$
6\exp\left(-\frac{k\Delta^2}{32M^2 L^4(4L^2 + 1)}\right) \leq \delta \Rightarrow k \geq \frac{32L^4 M^2(4L^2 + 1)}{\Delta^2}\log\left(\frac{6}{\delta}\right).
\tag{60}
$$

Finally, notice that $M = O(1/\epsilon^2)$, we obtain

$$
k = O\left(\frac{L^6}{\epsilon^4 \Delta^2}\log(1/\delta)\right)
\tag{61}
$$

## D. Equivalence conditions between S-Adam and the proximal method (Prox-SGD)

Let us define the proximal operator with a time-varying regularization coefficient $\gamma_t$ as follows

$$
x_{t+1} = \mathrm{prox}_{\eta_t h_t}(x_t) = \arg\min_x \left\{h_t(x) + \frac{1}{2\eta_t}\|x - x_t\|^2\right\},
\tag{62}
$$

where $h_t(x)$ is

$$
h_t(x) = \frac{\gamma_t}{2}\|x - x_t\|^2.
\tag{63}
$$

So we have the proximal algorithm iteratively solving the problem

$$
x_{t+1} = \arg\min_x \left\{(x - x_t)^T\left(\frac{m_t}{\sqrt{v_t} + \epsilon}\right) + \frac{1}{2\eta_t}\|x - x_t\|^2 + \frac{\gamma_t}{2}\|x - x_t\|^2\right\}.
\tag{64}
$$

Therefore, we immediately obtain the update formula for $x_t$

$$
x_{t+1} = x_t - \frac{\eta_t}{\eta_t\gamma_t + 1} \cdot \frac{m_t}{\sqrt{v_t} + \epsilon}.
\tag{65}
$$

When $\eta_t\gamma_t$ is very small, we have

$$
\frac{1}{\eta_t\gamma_t + 1} = (1 + \eta_t\gamma_t)^{-1} = 1 + (-\eta_t\gamma_t) + o((-\eta_t\gamma_t)^2) \approx 1 - \eta_t\gamma_t \approx e^{-\eta_t\gamma_t}.
\tag{66}
$$

If we let $\gamma_t = \frac{\lambda}{\eta_t}\rho_t$, then we have the update formula for S-Adam

$$
\begin{aligned}
x_{t+1} &\approx x_t - \eta_t e^{-\eta_t\left(\frac{\lambda}{\eta_t}\rho_t\right)} \cdot \frac{m_t}{\sqrt{v_t} + \epsilon} \\
&= x_t - \eta_t e^{-\lambda\rho_t} \cdot \frac{m_t}{\sqrt{v_t} + \epsilon}
\end{aligned}
\tag{67}
$$

where $\rho_t$ is the LGI estimate.

Therefore, under the first-order approximation, the exponential damping of S-Adam is equivalent to the fractional damping of the proximal method with dynamic regularization

$$
e^{-\lambda\rho_t} \approx \frac{1}{1 + \lambda\rho_t}.
\tag{68}
$$

# E. Relationship between S-Adam and stochastic smoothing methods

**Definition of stochastic smoothing.** For any non-smooth function $f : \mathbb{R}^d \to \mathbb{R}$, its stochastic smoothed version is defined as

$$f_\delta(x) = \mathbb{E}_{u \sim P}[f(x + \delta u)], \tag{69}$$

where $P$ is a uniform or Gaussian distribution, and $\delta > 0$ is the smoothing radius. Even if $f$ is a non-smooth function, then $f_\delta$ is infinitely differentiable, and we have

$$\nabla f_\delta(x) = \frac{d\mathbb{E}_u[f(x + \delta u)u]}{\delta}. \tag{70}$$

**Definition of Local Geometric Instability (LGI).** Our proposed LGI is defined as

$$\rho_t(x) = \frac{\mathrm{Var}_u\left[\frac{f(x+\delta u) - f(x)}{\delta}\right]}{\mathbb{E}\left[\left(\frac{f(x+\delta u) - f(x)}{\delta}\right)^2\right] + \epsilon} \tag{71}$$

For small $\delta$, Taylor expansion gives

$$Y(u) = \frac{f(x + \delta u) - f(x)}{\delta} = u^T \nabla f_\delta(x) + \frac{\delta}{2} u^T \nabla_\delta^2(x)u + o(\delta^2), \tag{72}$$

where $f_\delta$ is a smooth function with radius $\delta$. Note that even if $f$ is not smooth, $f_\delta$ is still smooth, and therefore it is legal to perform a Taylor expansion on $f_\delta$. To calculate LGI, we calculate the expectation and variance of $Y(u)$ separately.

**Calculation of the expectation and variance of $Y(u)$.** Let $g = \nabla f_\delta(x)$ and $H = \nabla^2 f_\delta(x)$, We calculate $\mathbb{E}[Y(u)]$ and obtain

$$\begin{aligned}
\mathbb{E}[Y(u)] &= \mathbb{E}[u^T g] + \frac{\delta}{2}\mathbb{E}[u^T H u] + o(\delta^2) \\
&= \frac{\delta}{2d}\mathrm{tr}(H) + o(\delta^2).
\end{aligned} \tag{73}$$

Note that $u$ is a random vector following a symmetric distribution (uniform or Gaussian distribution), and therefore we have $\mathbb{E}[u] = 0$ and $\mathbb{E}[u^T H u] = \frac{1}{d}\mathrm{tr}(H)$ (Wainwright, 2019).

Furthermore, we have

$$Y(u)^2 = (u^T g)^2 + \delta(u^T g)(u^T H u) + \frac{\delta^2}{4}(u^T H u) + o(\delta^2), \tag{74}$$

and calculate the expected value of each item

$$\begin{aligned}
\mathbb{E}[(g^T u)^2] &= \frac{1}{d}\|g\|^2, \\
\mathbb{E}[(g^T u)(u^T H u)] &= 0, \\
\mathbb{E}[(u^T H u)^2] &= \frac{(\mathrm{tr}(H))^2 + 2\|H\|_F^2}{d(d+2)},
\end{aligned} \tag{75}$$

So

$$\mathbb{E}[Y(u)^2] = \frac{1}{d}\|g\|^2 + \frac{\delta^2}{4d(d+2)}[(\mathrm{tr}(H))^2 + 2\|H\|_F^2] + o(\delta^2). \tag{76}$$

Finally, we obtain the variance of $Y(u)$ as

$$\begin{aligned}
\mathrm{Var}(Y(u)) &= \mathbb{E}[Y(u)^2] - (\mathbb{E}[Y(u)])^2 \\
&= \frac{1}{d}\|g\|^2 + \frac{\delta^2}{4d(d+2)}[(\mathrm{tr}(H))^2 + 2\|H\|_F^2] - \frac{\delta^2}{4d^2}(\mathrm{tr}(H))^2 + o(\delta^2) \\
&= \frac{1}{d}\|g\|^2 + \frac{\delta^2}{2d(d+2)}\|H\|_F^2 - \frac{\delta^2}{2d^2(d+2)}(\mathrm{tr}(H))^2 + o(\delta^2).
\end{aligned} \tag{77}$$

**LGI estimate.** Substituting the above estimates of mean and variance into LGI, we obtain

$$
\begin{aligned}
\rho_t(x) &= \frac{\mathrm{Var}(Y(u))}{\mathbb{E}[Y(u)^2] + \epsilon} \\
&= \frac{\frac{1}{d}\|g\|^2 + \frac{\delta^2}{2d(d+2)}\|H\|_F^2 - \frac{\delta^2}{2d^2(d+2)}(\mathrm{tr}(H))^2 + o(\delta^2)}{\frac{1}{d}\|g\|^2 + \frac{\delta^2}{4d(d+2)}[(\mathrm{tr}(H))^2 + 2\|H\|_F^2] + o(\delta^2) + \epsilon}
\end{aligned}
\tag{78}
$$

In high-dimensional optimization problems, especially when $d$ is large, the Hessian matrix often exhibits characteristics of a random matrix, hence: When $H$ is a random matrix, then

$$
\mathbb{E}[\mathrm{tr}(H)] = \sum_i \mathbb{E}[H_{ii}] = 0.
\tag{79}
$$

Furthermore, when $d$ is very large, then according to the law of large numbers, we have

$$
\frac{\mathrm{tr}(H)}{d} \to 0.
\tag{80}
$$

We obtain an approximate estimate of $\rho_t$ as

$$
\begin{aligned}
\rho_t(x) &\approx \frac{\frac{1}{d}\|g\|^2 + \frac{\delta^2}{2d(d+2)}\|H\|_F^2}{\frac{1}{d}\|g\|^2 + \frac{\delta^2}{2d(d+2)}\|H\|_F^2 + \epsilon} \\
&\approx \frac{\|g\|^2/d + \delta^2\|H\|_F^2/(2d^2)}{\|g\|^2/d + \delta^2\|H\|_F^2/(2d^2) + \epsilon} \\
&= \frac{1 + \frac{1}{2d}\kappa_\delta(x)^2}{1 + \frac{1}{2d}\kappa_\delta(x)^2 + \epsilon d/\|g\|^2},
\end{aligned}
\tag{81}
$$

where $g = \nabla f_\delta(x)$, $H = \nabla^2 f_\delta(x)$ and $\kappa_\delta(x)$ is the relative curvature of the function after random smoothing.

$$
\kappa_\delta(x) = \frac{\delta\|H\|_F}{\|g\|}.
\tag{82}
$$

**Conclusions.** When $\kappa_\delta \gg 0$ is high curvature/non-smooth, then $\rho \approx 1$; when $\kappa_\delta \approx 0$ is flat, then $\rho \approx 0$. Therefore, **LGI essentially measures the relative curvature of a function after random smoothing, and is a smoothness indicator.** Non-smooth regions correspond to high relative curvature, thus triggering damping.

## F. Stability comparison between S-Adam and Adam

**Empirical risk and expected risk.** For supervised learning, let the input space be $\mathcal{X}$, the output space be $\mathcal{Y}$, and the data distribution $\mathcal{D}$ be defined on $\mathcal{X} \times \mathcal{Y}$. We define a loss function $\ell(w; z)$ on the training set $S = \{z_1, \cdots, z_n\} \sim \mathcal{D}$ such that $z_i = (x_i, y_i)$

$$
\ell : \mathcal{W} \times (\mathcal{X} \times \mathcal{Y}) \to \mathbb{R},
\tag{83}
$$

where $\mathcal{W}$ is the parameter space. We have the following empirical risk and expected risk

$$
L_S(w) = \frac{1}{n}\sum_{i=1}^n \ell(w; z_i), \quad L_{\mathcal{D}}(w) = \mathbb{E}_{z \sim \mathcal{D}}[\ell(w; z)].
\tag{84}
$$

**Algorithm stability and generalization gap theorem.** Below we give the definition of uniform stability.

**Definition F.1.** (Uniform Stability). Algorithm $A : S \to w_S$ is $\beta$-uniform and stable if, for any datasets $S$ and $S'$ that are distinct only on one sample,

$$
\sup_z \mathbb{E}_A[|\ell(A(S); z) - \ell(A(S'); z)|] \le \beta,
\tag{85}
$$

where the expectation is based on the random sampling of the algorithm (random initialization, batch sampling).

We cite the following stability and generalization gap theorems without proof.

**Theorem F.2.** *(Stability and Generalization Gap,(Bousquet & Elisseeff, 2002),Theorem 12). If algorithm A is $\beta$-uniformly stable, then*

$$\mathbb{E}_S\mathbb{E}_A[L_\mathcal{D}(A(S)) - L_S(A(S))] \leq 2\beta, \tag{86}$$

*with a probability of at least $1 - \delta$,*

$$L_\mathcal{D}(A(S)) \leq L_S(A(S)) + 2\beta + (4n\beta + M)\sqrt{\frac{\ln(1/\delta)}{2n}}, \tag{87}$$

*where $M$ is the upper bound of the loss.*

Based on the above theorem, we can define a stability parameter $\beta$

$$\beta = \sup_{S,S'} \mathbb{E}_A[|\ell(w_T; z) - \ell(w'_T; z)|]. \tag{88}$$

Because the loss function $\ell$ is Lipschitz continuous, we have

$$\beta \leq L \cdot \mathbb{E}_A[\|w_T - w'_T\|]. \tag{89}$$

Below we compare the values of Adam and S-Adam on the function $\mathbb{E}_A[\|w_T - w'_T\|]$.

**Upper bound of the absolute value of the LGI difference.** We can define the directional gradient of the batch at step $t$ on the two datasets $S$ and $S'$

$$D_i^{(S)} = \frac{\ell(w_t + \delta u_i; S) - \ell(w_t; S)}{\delta}, \quad D_i^{(S')} = \frac{\ell(w_t + \delta u_i; S') - \ell(w_t; S')}{\delta}. \tag{90}$$

Furthermore, we set

$$\bar{D}^{(S)} = \frac{1}{k}\sum_{i=1}^{k} D_i^{(S)}, \qquad \bar{D}^{(S')} = \frac{1}{k}\sum_{i=1}^{k} D_i^{(S')} \tag{91}$$

$$\hat{V}^{(S)} = \frac{1}{k}\sum_{i=1}^{k}(D_i^{(S)} - \bar{D}^{(S)})^2 \qquad \hat{V}^{(S')} = \frac{1}{k}\sum_{i=1}^{k}(D_i^{(S')} - \bar{D}^{(S')})^2 \tag{92}$$

$$\hat{E}^{(S)} = \frac{1}{k}\sum_{i=1}^{k}(D_i^{(S)})^2 \qquad \hat{V}^{(S')} = \frac{1}{k}\sum_{i=1}^{k}(D_i^{(S')})^2 \tag{93}$$

**Lemma F.3.** *(Upper bound of the absolute value of the LGI difference). If we define the LGI as*

$$\rho_t(x) = \frac{\hat{V}^{(S)}}{\hat{E}^{(S)} + \epsilon}, \quad \rho'_t(x) = \frac{\hat{V}^{(S')}}{\hat{E}^{(S')} + \epsilon}, \tag{94}$$

*then we have $|\rho_t(x) - \rho'_t(x)| = O\left(\frac{1}{b}\right)$.*

*Proof.* To estimate $|\rho_t(x) - \rho'_t(x)|$, we will subsequently estimate $|D_i^{(S)} - D_i^{(S')}|$, $|\hat{E}^{(S)} - \hat{E}^{S'}|$, and $|\hat{V}^{(S)} - \hat{V}^{(S')}|$.

For a certain direction $u_i$, according to the Mean Value Theorem and $\|\ell(w; z)\| \leq L$, we have

$$\begin{aligned}
|D_i^{(S)} - D_i^{(S')}| &= \frac{1}{\delta b}|\ell(w + \delta u_i; z_k) - \ell(w_t; z_k) - (\ell(w_t + \delta u_i; z'_k) - \ell(w_t; z'_k))| \\
&= \frac{1}{\delta b}|\delta u_i^T \nabla\ell(\xi_i; z_k) - \delta u_i^T \nabla\ell(\xi'_i, z'_k)| \\
&\leq \frac{1}{b}\|\nabla\ell(\xi_i; z_k) - \nabla\ell(\xi'_i; z'_k)\| \\
&\leq \frac{2L}{b}.
\end{aligned} \tag{95}$$

According to the properties of convex functions, we have

$$
\begin{aligned}
|\hat{E}^{(S')} - \hat{E}^{(S)}| &\leq |(D^{(S)} - D^{(S')})^T \nabla \hat{E}| \\
&\leq 2 \sum_i |D_i^{(S)} - D_i^{(S')}||D_i^{(S)}| \\
&\leq 2 \max_i |D_i^{(S)} - D_i^{(S')}| \|D^{(S)}\|_\infty \\
&\leq \frac{4L}{b}\beta.
\end{aligned}
\tag{96}
$$

where $\|D^{(S)}\|_\infty = \beta$. Similarly, we have

$$
\begin{aligned}
|\hat{V}^{(S')} - \hat{V}^{(S)}| &\leq |(D^{(S)} - D^{(S')})^T \nabla \hat{E}| \\
&\leq 2 \sum_i |D_i^{(S)} - D_i^{(S')}||D_i^{(S)} - \bar{D}^{(S)}| \\
&\leq 2 \sum_i |D_i^{(S)} - D_i^{(S')}|(|D_i^{(S)}| + |\bar{D}^{(S)}|) \\
&\leq 4|D_i^{(S)} - D_i^{(S')}|\|D^{(S)}\|_\infty \\
&\leq \frac{8L}{b}\beta.
\end{aligned}
\tag{97}
$$

Ultimately, we have

$$
\begin{aligned}
|\rho_t(S') - \rho_t(S)| &\leq \|\nabla \rho_t(S)\| \|(\hat{V}^{(S')}, \hat{E}^{(S')}) - (\hat{V}^{(S)}, \hat{E}^{(S)})\| \\
&\leq \frac{1}{\epsilon}\sqrt{1 + \frac{(\hat{V}^{(S)})^2}{\epsilon^2}}\sqrt{|\hat{V}^{(S)} - \hat{V}^{S'}|^2 + |\hat{E}^{(S)} - \hat{E}^{(S')}|^2} \\
&\leq \frac{1}{\epsilon}\sqrt{1 + \frac{L^2}{\epsilon^2}}\frac{4L\beta}{b}\sqrt{5} = O\left(\frac{1}{b}\right).
\end{aligned}
\tag{98}
$$

$\square$

**Boundary of parameter difference norm.** Consider two datasets,

$$
S = \{z_1, \cdots, z_n\}, \quad S' = \{z_1', \cdots, z_n'\},
\tag{99}
$$

where they differ only in the $k$-th sample $z_k \neq z_k'$, and otherwise $z_i = z_i'$ for all $i \neq k$. Let the batch index set be $I_t \subseteq \{1, \cdots, n\}$, with size $|I_t| = b$. Therefore, we can define the update of the parameter at step $t$ on the two datasets $S$ and $S'$

$$
w_{t+1} = w_t - \eta_t d_t, \quad w_{t+1}' = w_t' - \eta_t d_t'.
\tag{100}
$$

Furthermore, we have

$$
\|w_{t+1} - w_{t+1}'\| \leq \|w_t - w_t'\| + \eta_t \|d_t - d_t'\|.
\tag{101}
$$

Comparing Adam and S-Adam, we obtain the following two different update directions

$$
d_t^{\text{Adam}} = g_t = \frac{m_t}{\sqrt{v_t} + \epsilon}, \quad d_t^{\text{S-Adam}} = \alpha_t g_t = e^{-\lambda \rho_t}\frac{m_t}{\sqrt{v_t} + \epsilon}.
\tag{102}
$$

We estimate the upper bound of $\|d_t^{\text{S-Adam}} - d_t'^{\text{S-Adam}}\|$

$$
\begin{aligned}
\|d_t^{\text{S-Adam}} - d_t'^{\text{S-Adam}}\| &= \|\alpha_t g_t - \alpha_t' g_t'\| \\
&= \|\alpha_t g_t - \alpha_t g_t' + \alpha_t g_t' - \alpha_t' g_t'\| \\
&\leq \alpha_t \|g_t - g_t'\| + |\alpha_t - \alpha_t'|\|g_t'\| \\
&= \alpha_t \|d_t^{\text{Adam}} - d_t'^{\text{Adam}}\| + |\alpha_t - \alpha_t'|\|g_t'\|.
\end{aligned}
\tag{103}
$$

Notice that $\|g_t\| \le L'$ and Lemma F.3, then we have ($e^{-x}$ is a convex function)

$$\|g'_t\||\alpha_t - \alpha'_t| = |e^{\lambda\rho_t} - e^{\lambda\rho'_t}| \le |\lambda e^{-\lambda\rho'_t}(\rho_t - \rho'_t)| \le \lambda|\rho_t - \rho'_t| \le \frac{C\lambda L}{b}. \tag{104}$$

When the batch size $b$ is large, the second term can be ignored, that is

$$\|d_t^{\text{S-Adam}} - d'^{\text{S-Adam}}_t\| \le \alpha_t\|d_t^{\text{Adam}} - d'^{\text{Adam}}_t\|. \tag{105}$$

According to Eqn. (101), we have

$$\|w_T - w'_T\| \le \sum_{t=1}^{T} \eta_t\|d_t - d'_t\| \tag{106}$$

Therefore, we obtain $\beta_{\text{Adam}}$

$$\beta_{\text{Adam}} \le L\sum_{t=1}^{T} \eta_t\|d_t^{\text{Adam}} - d'^{\text{Adam}}_t\| = C_{\text{Adam}}. \tag{107}$$

Futhermore, we have

$$
\begin{aligned}
\beta_{\text{S-Adam}} &= L\sum_{t=1}^{T} \eta_t\|d_t^{\text{S-Adam}} - d'^{\text{S-Adam}}_t\| \\
&\le L\sum_{t=1}^{T} \eta_t\alpha_t\|d_t^{\text{Adam}} - d'^{\text{Adam}}_t\| \\
&\le (\max_t \alpha_t)L\sum_{t=1}^{T} \eta_t\|d_t^{\text{Adam}} - d'^{\text{Adam}}_t\| = (\max_t \alpha_t)C_{\text{Adam}}.
\end{aligned}
\tag{108}
$$

For non-smooth regions, $\rho_t$ becomes relatively high, so $\alpha_t = e^{-\lambda\rho_t}$ is reduced. Thus S-Adam dampens unstable updates more strongly than Adam in geometrically unstable regions.

## G. Additional Experimental Details

### G.1. Experimental Hyperparameters

*Table 6.* Expanded Hyperparameters for Experiment 1

| Parameter | AdamW | Prox-SGD | S-Adam (Ours) |
|---|---|---|---|
| Learning Rate ($\eta$) | 0.001 | 0.01 | 0.001 |
| Weight Decay ($\lambda_{wd}$) | 0.01 | – | 0.01 |
| L1 Regularization ($\lambda_{L1}$) | – | 0.0001 | – |
| Momentum / Betas | (0.9, 0.999) | 0.9 | (0.9, 0.999) |
| Batch Size | 128 | 128 | 128 |
| Epochs | 10 | 10 | 10 |
| *S-Adam Specific Geometry Params* | | | |
| Perturbation Scale ($\delta$) | – | – | 0.01 |
| Directional Probes ($K$) | – | – | 2, 8 |
| Damping Intensity ($\lambda_{LGI}$) | – | – | 2.0 |
| LGI Score Cap | – | – | 10.0 |
| Stabilization $\epsilon$ | – | – | $10^{-6}$ |

*Table 7.* Expanded Hyperparameters for Experiment 2

| Parameter | AdamW | Prox-SGD | S-Adam (Ours) |
|---|---|---|---|
| Learning Rate ($\eta$) | 0.001 | 0.01 | 0.001 |
| Weight Decay ($\lambda_{wd}$) | 0.01 | – | 0.01 |
| L1 Regularization ($\lambda_{L1}$) | – | 0.0001 | – |
| Momentum / Betas | (0.9, 0.999) | 0.9 | (0.9, 0.999) |
| Batch Size | 2,4,16,64 | 2,4,16,64 | 2,4,16,64 |
| Epochs | 20 | 20 | 20 |
| *S-Adam Specific Geometry Params* | | | |
| Perturbation Scale ($\delta$) | – | – | 0.01 |
| Directional Probes ($K$) | – | – | 2, 8 |
| Damping Intensity ($\lambda_{LGI}$) | – | – | 2.0 |
| LGI Score Cap | – | – | 10.0 |
| Stabilization $\epsilon$ | – | – | $10^{-6}$ |

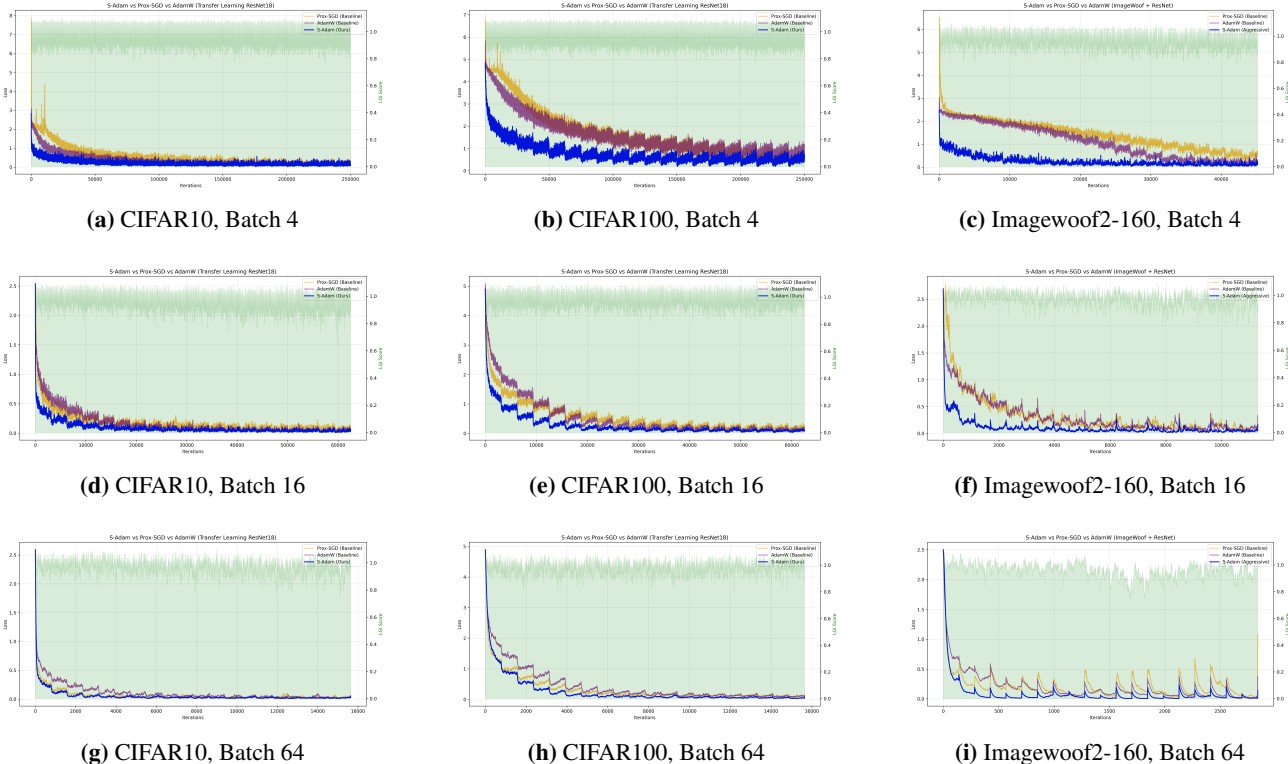

**(a)** CIFAR10, Batch 4   **(b)** CIFAR100, Batch 4   **(c)** Imagewoof2-160, Batch 4

**(d)** CIFAR10, Batch 16   **(e)** CIFAR100, Batch 16   **(f)** Imagewoof2-160, Batch 16

**(g)** CIFAR10, Batch 64   **(h)** CIFAR100, Batch 64   **(i)** Imagewoof2-160, Batch 64

*Figure 6.* Loss curves for ResNet18 across CIFAR10, CIFAR100, and Imagewoof2-160 datasets with varying batch sizes (4, 16, and 64).

