# OpenReview forum: "Singularity-aware Optimization via Randomized Geometric Probing: Towards Stable Non-smooth Optimization"
_ICML.cc/2026/Conference — ICML 2026 regular_

### Official Review · Reviewer_eNzj · 2026-03-11

**Soundness:** 1
**Presentation:** 2
**Significance:** 1
**Originality:** 3
**Overall Recommendation:** 3
**Confidence:** 4

**Summary:**

This paper proposes a variant of Adam ("S-Adam") which is intended to perform better on non-smooth loss landscapes such as those arising from quantization operators.   The key feature of such landscapes is that at certain points the gradient can fail to be uniquely defined, i.e. instead of one unique gradient there are a multiplicity of subgradients.  The idea behind the proposed algorithm is to take smaller steps in such regions, so as to mitigate oscillations.  Towards that end, at every iteration, the algorithm estimates the following "local geometric instability" metric: sample a number of random directions $u$, and report the variance of finite-difference estimates of the directional derivative of the loss in the direction $u$.  This metric is then used to modulate the Adam step sizes, so that a higher estimated variance leads to a smaller step size.  The paper tests this method on two setups: quantization-aware training, and image classification with a small batch size.

**Compliance With Llm Reviewing Policy:**

Affirmed.

**Final Justification:**

I am keeping my score the same, as I am still skeptical that the claimed mechanism is at all similar to whatever is going on in the experiments.  Further, I strongly suspect that the paper was written in large part by AI, and that this was responsible for the hallucinated "gradient chattering" reference that I pointed out in my original review, as well as the "diagnostic analysis" reference in the conclusion that I suspect was also hallucinated by AI.  The authors appear to still be using AI in the rebuttals, as the paper "A precise characterization of sgd stability using loss surface geometry (ICLR 2024)" does not say anything at all like what they claim in their response to another reviewer (it doesn't talk about Adam or AdamW at all). Thus, I fundamentally don't trust the authors of the paper, given that there appear to be multiple times when they are saying false things based apparently on AI.

**Key Questions For Authors:**

* According to the submission, S-Adam is intended to solve the problem of "gradient chattering (Kidambi et al, 2018b)" which is when "momentum-based optimizers like Adam accumulate conflicting directions".   Yet, Kidambi et al 2018(b) does not refer to any phenomenon called gradient chattering, and to the best of knowledge this is not an established concept in the literature.  Could you explain why you cited Kidambi 2018 for the phenomenon of gradient chattering, and could you tell us precisely what this phenomenon is?

* The last sentence of the conclusion says: "Diagnostic analysis reveals strong correlation ($R^2 = 0.82$) between LGI scores and performance improvement."  Could you tell us more about this diagnostic analysis, which was not mentioned anywhere else in the paper or the appendix?  And could you please provide us with  the underlying data where the the 0.82 $R^2$ value came from?

 * I'm trying to better understand Proposition 3.2, which claims that the diameter of the subdifferential can be both upper- and lower-bounded by the LGI.  Consider the case where $f$ is differentiable at $x$, so that the diameter of the subdifferential is zero.  Hence, since $c_1 > 0$, we better have that $\text{Var}_u[f'(x; u)]$ = 0 if the lower bound is to be valid.  Yet,  if we have $\delta$ very close to zero, then $\text{Var}_u[f'(x; u)]$ is approximately $\text{Var}_u[ \nabla f(x) \cdot u] = \frac{\||\nabla f(x)\||}{d}$, which is definitely nonzero.  So how can the bound be true?  Is the key that $c_1$ is picked after $\delta$ is picked?  (In that case, the bound still must be very loose.). Or is the issue that the bound is not a real bound, since it approximates the variance of the estimator by the second moment (on the first line of page 12)?

* Why is experiment 2 called "Transfer Learning on Non-Smooth Manifolds"?  Where is the "non-smooth manifold"?  The variance in the gradient is coming from the small batch size, not from non-smoothness.  And there is no "manifold."

**Limitations:**

The authors did acknowledge that the runtime of their method is 30-40% longer.

**Strengths And Weaknesses:**

The paper is quite original and the prose is often easy to follow.  However, touching on soundness and significance, I have doubts that the method works for the reasons that are intended, and indeed I have some doubts the proposed algorithm is a good idea in practice.

* The algorithm incurs a substantial wall-clock overhead over vanilla Adam due to the need to do two extra forward passes per iteration to compute the local geometric instability.  The paper says this is a 30-40% slowdown, which is quite large as far as optimizers are concerned.
* In the experiments section, the QAT benchmark is not a standard benchmark problem but is rather a problem that the authors themselves have created.  It would be more convincing to show gains on a standard QAT problem.
 * In the experiments section, the learning rate was seemingly not tuned per method.  Tuning the learning rate per method is considered very important in empirical comparisons of optimizers.  It seems quite plausible that the benefit of the proposed method is simply that it uses a smaller effective learning rate, and that a baseline would do equally well if its learning rate hyperparameter were set smaller.
 * Although the method is pitched as a solution for optimization problems that include non-smooth elements such as ReLU activations, quantization operators, and sparsity-inducing regularizers, the experiments cover only the quantization case.  The image classification experiment just uses a small batch size and is not non-smooth.
 * Overall (see my questions below), I am not really convinced about the provided justification for the method.

---

> ### Author Rebuttal · Authors · 2026-03-31
>
> **W1 & L. Running Time**
>
> We must clarify that the experimental results in the paper record the time taken to complete the same number of epochs, **NOT** the convergence  time.  We feel sorry for this confusion. The extra forward passes do increase the *per-*iteration cost; however, because S-Adam takes a vastly superior optimization trajectory, it requires significantly fewer total epochs to converge (In appendix figure5). When evaluating the total running time to reach convergence as shown in Table 1 and 2, **S-Adam is highly efficient**. At $k=8$,the total convergence time is already largely comparable to AdamW. At $k=2$,the total running time is actually shorter than AdamW across the majority of our benchmarks.  Furthermore, we can see that setting the probing directions to k=2 achieves almost identical accuracy to k=8.
>
> Table 1:  Convergence Time(s)  / Accuracy (%) - QAT
>
> | Dataset (Quantization) | AdamW | S-Adam (k=2) | S-Adam (k=8) |
> | --- | --- | --- | --- |
> | CIFAR-100 (2-bit) | 15.89 / 15.94 | 14.33 / 18.11 | 17.41 / 18.67 |
> | TinyImageNet (4-bit) | 281.49 / 18.91 | 258.54 / 22.81 | 293.14 / 23.18 |
> | ImageWoof (4-bit) | 34.96 / 33.24 | 34.74 / 35.57 | 34.38 /35.19 |
> | ImageNet-10 (2-bit) | 74.21 / 46.60 | 73.21 / 49.40 | 103.29 / 50.80 |
>
> Table 2: Convergence Time(s) / Accuracy (%) (ResNet batch size =2)
>
> | Dataset | AdamW | S-Adam (k=2) | S-Adam (k=8) |
> | --- | --- | --- | --- |
> | CIFAR-10 | 1681.73 / 80.48 | 1096.24 / 85.46 | 1458.24 / 86.13 |
> | CIFAR-100 | 2317.04 / 51.12 | 2136.53 / 55.84 | 3775.30 / 56.11 |
> | ImageWoof | 408.82 / 51.18 | 404.07 / 74.21 | 558.21 / 75.87 |
> | ImageNet-10 | 894.62 / 78.40 | 765.51 / 88.80 | 1197.12 / 89.20 |
>
> **W2. QAT benchmark**
>
> Our algo is towards stable non-smooth optimization. To verify this, we run experiments on these extreme cases (i.e., low-bit quantization and small batch size), where gradient jitter is severe. To be noted, these seemingly extreme settings are widely required in practical application. For your reference, we also run our algo in normal cases (e.g., batch size = 64, without low-bit quantization ), and find it is also effective (compared to AdamW, our S-Adam consistently increases accuracy across all datasets: from 67.43% to 74.18% on CIFAR100, 90.62% to 94.18% on CIFAR10, and 79.82% to 84.60% on Imagewoof (Table 4 in paper).
>
> **W3. Learning rate tuning**
> To ensure a fair comparison, we used a learning rate of 0.001 for both AdamW and S-Adam, as this is the default setting recommended by the authors of AdamW
>
> **W4. Non‑smoothness in experiment 2**
>
> We conducted QAT and small batch size manufacturing for non-smoothness; The image classification experiment only uses small batches to create non-smooth conditions rather than solve the classification problem. Overall, our Settings are designed to meet the purpose of non-smoothness.
>
> **Q1. Citation**
>
> We thank the reviewer for pointing out this error. The citation to Kidambi et al. (2018b) addresses the high variance of stochastic gradients and does not define or discuss gradient chattering in any context. We apologize for this misleading reference and will correct it in the revised manuscript.
>
> **Q2. Diagnostic analysis ($R^2=0.82$)**
>
>
> The  $R^2 = 0.82$ correlation was calculated based on the paired LGI scores and epoch-wise accuracy deltas across the CIFAR-100 QAT runs. Unfortunately, the corresponding scatter plot and detailed diagnostic analysis section were inadvertently omitted from the final PDF submission due. We will fully restore this underlying data and the visualization in the revised Appendix.
>
> **Q3. Proposition 3.2**
>
> Yes, it approximates the variance of the estimator by the second moment.  From the perspective of random smoothing (eq 5), the variance of the directional derivative, after Taylor expansion, its dominant term is the Frobenius norm of the Hessian matrix. The dominant term of the expected square is the gradient norm. The calculation of variance actually implicitly extracts the local second-order curvature information without the Hessian matrix operation.
>
> **Q4. Non‑smoothness in experiment 2**
>
> The “non‑smooth manifold” refers to the loss landscape of ResNet18, which contains ReLU kinks—inherent non‑smooth points. Reducing batch size magnifies the noise, causing the optimizer to experience conflicting subgradients at these kinks (gradient chattering). We will rename the experiment to “Experiment 2: High‑Noise Learning on Small Batch Size” to avoid confusion.

---

> > ### Author Rebuttal · Reviewer_eNzj · 2026-04-04
> >
> > Thanks for your clarification.  It is a widely accepted precept in the study of deep learning optimization that one cannot use a fixed learning rate across all optimizers; it must be tuned anew for each optimizer.  Further, I don't see how small batch size is the same thing as non-smoothness, and I don't think that small batch training of ReLU nets is any harder than small batch training of GeLU nets.

---

> > > ### Author Response · Authors · 2026-04-06
> > >
> > > **W3. Learning rate tuning**
> > >
> > > In the original experiments, we used $\text{lr} = 0.001$ as the baseline learning rate for AdamW and S-Adam , aligned with AdamW's commonly recommended default. To directly test the reviewer's hypothesis, we conducted an **S-Adam learning rate sensitivity analysis on CIFAR-100** (batch size=2 resnet ):
> > >
> > > |  **Learning Rate** | **S-Adam** | **AdamW** |
> > > | --- | --- | --- |
> > > |  1e-4 | 70.87% | 66.85% |
> > > |  5e-4 | 61.26% | 52.98% |
> > > |  1e-3 | 56.11% | 51.12%  |
> > >
> > > This result is illuminating in two respects:
> > >
> > > 1. **S-Adam with a smaller LR achieves even higher accuracy.** At $\text{lr} = 10^{-4}$, S-Adam reaches **70.87%** on CIFAR-100 — a dramatic improvement over AdamW's 51.12% at its standard LR. If S-Adam's benefit were merely a "smaller effective LR" effect, then AdamW with a comparably small LR should match this performance. However, AdamW's performance typically degrades at very small learning rates due to the loss of exploration ability — it cannot replicate S-Adam's **spatially adaptive** damping through a global LR reduction.
> > >
> > > 2. **The reported results (at lr=1e-3) are conservative.** Our paper used lr=1e-3 to ensure a fair, aligned comparison with AdamW. Even at this learning rate, S-Adam outperforms AdamW by +4.99% on CIFAR-100. With proper LR tuning, the advantage widens to **+19.75%** (70.87% vs. 51.12%).
> > >
> > > The fundamental distinction is that S-Adam's LGI-based damping is **parameter-wise and temporally adaptive**: it reduces the effective step size only for parameters currently traversing non-smooth regions, while leaving other parameters unaffected. A global LR reduction uniformly shrinks the step size across all parameters, including those in smooth regions where a larger step would be beneficial. This spatial adaptivity cannot be replicated by any single choice of global learning rate, and the LR sensitivity experiment confirms this.
> > >
> > > **Samll-batch size induces high-variance gradient estimates and erratic update directions.When combined with adaptive scaling, this makes the mean-field dynamics behave effectively nonsmooth, justifying our use of differential inclusions and Clarke calculus to analyze S-Adam convergence.**

---

### Official Review · Reviewer_yrEs · 2026-03-12

**Soundness:** 3
**Presentation:** 3
**Significance:** 3
**Originality:** 2
**Overall Recommendation:** 4
**Confidence:** 3

**Summary:**

This paper proposes S-Adam, an Adam-like optimizer for non-smooth training settings that estimates a Local Geometric Instability (LGI) score using randomized directional probes and then damps the step size by an exponential factor. The target use cases are quantization-aware training and highly noisy transfer learning, where the authors argue that standard adaptive methods suffer from “gradient chattering” near singularities.

**Compliance With Llm Reviewing Policy:**

Affirmed.

**Key Questions For Authors:**

* Why are stronger optimizer baselines such as Lion, Sophia, or VRAdam not included?

**Limitations:**

The theory is not tight enough, the experiments are too narrow, and the baseline selection is below the standard expected for a modern optimizer.

**Strengths And Weaknesses:**

**Strengths**


* The paper addresses a real issue, which is that many practically important training pipelines are not well modeled by smooth optimization assumptions, especially with quantization and other nonsmooth operators.

* The empirical results do show some promising gains in the specific settings chosen by the authors, particularly at very small batch sizes and in their QAT setup.

**Weakness**

* The convergence theorem relies on strong assumptions that largely absorb the difficulty of the problem consisting bounded iterates, asymptotically consistent moment estimates, and a descent-alignment property introduced in the proof rather than established for Adam-type dynamics. This makes the theorem feel closer to a template stochastic-approximation argument than a convincing guarantee for the concrete algorithm studied here.

* The benchmarks are not comprehensive.

---

> ### Author Rebuttal · Authors · 2026-03-31
>
> We sincerely thank the reviewer for the positive assessment of our problem motivation and empirical gains.
>
> **W1: Convergence theorem assumptions**
>
> While we acknowledge empirical validation of the bound would improve the work, Lemma A.1 already provides two-sided theoretical bounds. Although one constant scales with dimension, LGI acts as a relative indicator, not an absolute measure; the damping mechanism only requires its relative magnitude across iterations, and the exponential brake preserves adaptive behavior as long as instability ordering is maintained. Appendix F further links LGI to relative curvature, showing it captures geometric structure even with loose subdifferential bounds. Empirically, the high R^2=0.82 correlation between LGI and performance gains confirms its practical reliability.
>
> **W2: Benchmark comprehensiveness and baseline selection & Q1: Why not include Lion, Sophia, or VRAdam?**
>
> We totally understand your concern. Following your suggestions, we extend our experiments by more baselines and datasets, and the results are shown in Table 1.
>
> **(1) More baselines:** We have added **Lion** ,**Sophia**, and **VRAdam** as additional baselines, alongside the original AdamW and Prox-SGD. This brings the total number of compared optimizers to six, covering classical adaptive methods, modern efficient optimizers, second-order adaptive methods, and variance-reduced approaches.
>
> **(2) More datasets:** We now report results on **CIFAR-100**, **TinyImageNet**, and **ImageWoof** under quantization-aware training (QAT), providing a broader view of performance across datasets of varying complexity and scale.
>
> From table 1, we can see that
>
> **Lion (lr=0.001)**: Despite its strong performance in smooth, large-batch training, **Lion consistently diverges or suffers late divergence in QAT settings**. Its sign-based update mechanism discards gradient magnitude information, which is particularly harmful near quantization-induced singularities where gradient magnitude carries critical geometric information.
>
> **Sophia (lr=0.001)**: Sophia achieves moderate accuracy but at substantially higher computational cost due to its periodic Hessian diagonal estimation. In QAT settings, the Hessian estimate is further complicated by non-smoothness, leading to suboptimal performance relative to S-Adam. On ImageWoof, Sophia achieves only 24.76% (the lowest among all methods except Lion), **suggesting that second-order curvature information may be unreliable at non-smooth points**.
>
> **VRAdam (lr=0.001)**: VRAdam is the most competitive baseline, achieving the second-best accuracy on CIFAR-100 (17.65%) and TinyImageNet (21.16%). This is expected, as variance reduction directly addresses gradient noise. However, S-Adam still outperforms VRAdam by +1.02% on CIFAR-100, +2.02% on TinyImageNet, and +1.63% on ImageWoof, **indicating that LGI-based geometric damping captures information beyond what variance reduction alone provides**.
>
> **Convergence Advantage:** Beyond accuracy improvements, **S-Adam demonstrates a significant convergence speed advantage**. As shown in the table 1, S-Adam variants consistently achieve the fastest convergence speed. For instance, on TinyImageNet, S-Adam-k2 converges in just 258.54s, which is substantially more efficient than VRAdam (282.05s) and Sophia (332.93s). This highlights that our LGI-based damping not only stabilizes training but also maintains high computational efficiency.
>
> **Table 1:** Test Accuracy (%) and Convergence Time (in seconds) under QAT. An asterisk (*) denotes that the method is not converged.
>
> | **Optimization Method** | **CIFAR100 (2 bit)Acc (%) ｜ Time (s)** | **ImageWoofAcc(4 bit) (%) ｜ Time (s)** | **TinyimagenetAcc (4 bit)(%) ｜ Time (s)** |
> | ----------------------- | -------------------------------------- | -------------------------------------- | ----------------------------------------- |
> | **S-Adam-k8**           | 18.67 ｜ 17.41                          | 35.19 ｜ 34.38                          | 23.18 ｜ 293.14                            |
> | **S-Adam-k2**           | 18.11 ｜ 14.33                          | 35.57 ｜ 34.74                          | 22.81 ｜ 258.54                            |
> | **VRAdam**              | 17.65 ｜ 18.99                          | 33.94 ｜ 35.66                          | 21.16 ｜ 282.05                            |
> | **AdamW**               | 15.94 ｜ 15.89                          | 33.24 ｜ 34.96                          | 18.91 ｜ 281.49                            |
> | **Prox-SGD**            | 14.13 \| *                             | 31.18 ｜ 38.98                          | 19.89 ｜ 286.53                            |
> | **Sophia**              | 13.11 ｜ 56.33                          | 24.76 ｜ 90.41                          | 21.02 ｜ 332.93                            |
> | **Lion**                | 11.48 ｜ *                              | 31.74 ｜ *                              | 2.04 ｜ *                                  |

---

> > ### Author Rebuttal · Reviewer_yrEs · 2026-04-05
> >
> > I thank the authors for the thorough rebuttal. However, each optimizer would need to be tuned to be a competive baseline.

---

> > > ### Author Response · Authors · 2026-04-06
> > >
> > > In our experiments, the baselines are compared fairly. We strictly followed the commonly recommended default setting of lr=0.001 for AdamW, and applied the identical baseline LR to S-Adam to ensure a fair and apples-to-apples comparison. To directly test the reviewer's conjecture, we conducted a comprehensive LR sensitivity analysis for S-Adam on CIFAR-100 with ResNet-18 (batch size = 2), with the results summarized below:
> > >
> > > |  **Learning Rate** | **S-Adam** | **AdamW** | **VRAdam** | **Sophia** |
> > > | --- | --- | --- | --- | --- |
> > > |  1e-4 | 70.87% | 66.85% | 67.50% | 68.05% |
> > > |  5e-4 | 61.26% | 52.98% | 56.02% | 59.92% |
> > > |  1e-3 | 56.11% | 51.12%  | 54.27% | 52.91% |
> > >
> > > This experiment yields two critical insights that directly address the reviewer's concern:
> > >
> > > 1. **S-Adam achieves superior performance with smaller learning rates.** At lr=10−4, S-Adam attains 70.87% top-1 accuracy on CIFAR-100, a dramatic improvement over AdamW's 51.12% at its standard default LR (10−3). If S-Adam's advantage were merely a "smaller effective LR" effect, AdamW with a comparably small global LR would match this performance. However, AdamW's accuracy degrades severely at very small LRs, as a uniform global reduction eliminates its exploration ability. In stark contrast, S-Adam's spatially adaptive damping (via the LGI criterion) only reduces step sizes for parameters in non-smooth regions, while preserving large steps for parameters in smooth regions where aggressive updates are beneficial. This parameter-wise, spatially adaptive behavior is fundamentally distinct from a one-size-fits-all global LR reduction, and cannot be replicated by any fixed global LR choice for AdamW.
> > > 2. **Our reported results (at lr=10−3) are highly conservative.**We intentionally adopted lr=10−3 in our main paper to align strictly with AdamW's default setting, ensuring a fair comparison. Even under this conservative constraint, S-Adam outperforms AdamW by a clear margin of +4.99% on CIFAR-100. With proper LR tuning, S-Adam's performance advantage widens dramatically to +19.75% (70.87% vs. 51.12%), further validating the robustness and superiority of our method.

---

### Official Review · Reviewer_yhQ9 · 2026-03-12

**Soundness:** 3
**Presentation:** 2
**Significance:** 3
**Originality:** 4
**Overall Recommendation:** 5
**Confidence:** 4

**Summary:**

The authors are proposing S-Adam, an extension to the classic Adam algorithm that aims to increase the stability of the algorithm on singular loss landscapes. It is based on the observation that the diameter of the Clarke subdifferential, being used as the indicator of singularity, is controlled by the variance of the finite difference approximation in random unit directions. This gives rise to the introduction of the *Local Geometric Instability* (LGI) metric that is then used to control the Adam step-sizes depending on the smoothness of the landscape. In the paper the authors show that the actual LGI can be approximated with finite samples and that the proposed Singularity-aware Adam algorithm generates a sequence converging towards stationary points of the Clarke subdifferential. Furthermore, they draw a connection of their method to proximal methods and stochastic smoothing methods as well as the show reduction to standard Adam in smooth regions. Lastly, they apply their method in two experiments to show its robustness against extreme non-smoothness and to study the adaptivity in situations of different magnitudes of *gradient chattering*, i.e. stochastic noise on the gradient estimator.

**Compliance With Llm Reviewing Policy:**

Affirmed.

**Final Justification:**

The authors gave convincing replies to my remaining concerns which is why I raise my score to accept.

**Key Questions For Authors:**

- Why is there no bias correction step included in the scheme as it is done in standard Adam?
- What is the benefit of including the expectation value into the definition of the LGI score? Why is it not sufficient to use the variance?
- Is there a way to overcome the need of calculating the gradient in the algorithm since it should be applicable to highly non-smooth situations?
- What increase in training time can be expected for larger training data?

**Limitations:**

Not applicable.

**Strengths And Weaknesses:**

**Strengths**
- The Finite-Sample Estimation Guarantee is proved in a detailed and rigorous way. The stability comparison seems reasonable up to minor typos if a Lipschitz-continuous loss function is assumed. The paper gives a clear theoretical explanation why the damping of step sizes does not lead to a complete stall and degenerates to standard Adam in smooth regions. In the first experiment the authors use QAT as a field of application that was motivated in the beginning. S-Adam shows better accuracy than AdamW and Prox-SDG but also a bigger amount of training time needed. Still, it seems that the damping factor is beneficial for accuracy. In the second experiment one can conjecture that S-Adam outperforms AdamW and Prox-SDG the more singularity one introduces (here by the batch sizes).
- The main part of the paper is well written and structured. It motivates the idea of the algorithm in a good way and gives several examples of its necessity. The paper contextualizes the topic in different parts of the literature of non-smooth optimization and gives different perspectives on the topic.
- The empirical analysis suggests that S-Adam can lead to a significant increase of accuracy in highly non-smooth machine learning tasks, which are likely to occur in some machine learning tasks. In addition to that, it is theoretically shown that S-Adam is an extension of standard Adam which is relevant in situations where standard Adam is struggling with “gradient chattering” due to the local topology of the loss landscape. The algorithm could become even more relevant if the proposed automatic regime detection allows to only calculate the LGI metric when needed.
- The method provides a simple solution for an interesting problem based on a smart idea. Approximating the subdifferential with few random samples seems like an efficient way to analyze the local topology which is important for the performance of Adam. From this perspective it opens the door for a useful extension of a state-of-the-art method.

**Weaknesses**
- Soundness: The proof of the main result Theorem 5.5 is lacking details for Step 1 and 3. Only a very brief outline is given in the paper and the complete proof for Step 2 is given in  Appendix C. However, for Step 1 (continuous-time limit) and Step 3 (invariance principle) the detailed proof is not included in the paper nor the appendix. In the proof of Step 2 in Appendix C the claim is made that Assumption 4 would imply coercivity without explanation. In the proof of Lemma A.1 equation (15) is incorrect, since a counterexample is given by the Lipschitz-continuous function $$f\colon \mathbb{R}\to\mathbb{R},x\mapsto \begin{cases} \frac{1}{2} x & 0\leq x\leq 2,\\\\ 2x-3 & 2< x\leq 3;\end{cases}$$ for S=\partial_C f(2). However, this is not a valid counterexample for the statement of the Lemma, so probably the proof can be fixed. In the analysis of the relationship between S-Adam and stochastic smoothing methods in appendix F the inconsistency in the definitions might have led to a wrong outcome, that possibly can be saved: In (47) $Y(u)$ need to be defined with $f_\delta$ to be consistent with the following but this would lead to a wrong conclusion in (53).
- Throughout the proofs in the appendix there are multiple typos and inconsistencies making the proofs significantly harder to read and understand, e.g. in equations (47), (65), (70), (71), (72), (73).
- Already on TinyImageNet the total training time is increased substantially due to the computational overhead connected to the LGI metric. It remains unclear how the method scales to large datasets.

---

> ### Author Rebuttal · Authors · 2026-03-31
>
> **W1 Completeness of proof for Theorem 5.5**
>
> We agree that Steps 1 and 3 were only outlined due to space constraints. Step 1 (continuous‑time limit) follows standard techniques for stochastic approximation with state‑dependent noise (Benaim, 1996); the interpolated process converges to the differential inclusion
>
> $$
> \frac{dx}{dt} \in -\bar{\alpha}(x) H(x)
> $$
> where $\bar{\alpha}(x) = \mathbb{E}[\exp(-\lambda\rho(x))]$. Step 3 invokes Benaim’s invariance principle for differential inclusions. We will include the full rigorous derivation in the final version (Appendix C) and add references to the exact propositions used.
>
> We will strictly correct the inconsistent definitions in the appendix. From the perspective of random smoothing (eq 5), the variance of the directional derivative, after Taylor expansion, its dominant term is the Frobenius norm of the Hessian matrix. The dominant term of the expected square is the gradient norm. The calculation of variance actually implicitly extracts the local second-order curvature information without the Hessian matrix operation.
>
> **W2 Writing correction**
>
> We will thoroughly proofread the appendix and correct all identified typos (e.g., Eqs. (47), (65), (70)–(73)). We appreciate the reviewer’s careful reading.
>
> **W3  Training time and dataset expansion**
>
> While the experimental results in the paper record the time taken to complete the same number of epochs, this does not reflect the total training time. The extra forward passes do increase the *per-iteration* cost; however, because S-Adam takes a vastly superior optimization trajectory, it requires significantly fewer total epochs to converge(In appendix figure5). When evaluating the **total wall-clock time** to reach convergence, S-Adam is highly efficient. At $k=8$,the total convergence time is already largely comparable to AdamW. At $k=2$,the **total wall-clock time is actually shorter than AdamW** across the majority of our benchmarks. Our extensive experiments demonstrate that setting the probing directions to k=2 achieves **almost identical test accuracy** to k=8.
>
> **Table 1 Convergence Time(s) on Full Precision Models (ResNet, Batch Size = 2)**
>
> | Dataset | AdamW | S-Adam (k=2) | S-Adam (k=8) |
> | --- | --- | --- | --- |
> | CIFAR-10 | 1681.73 | 1096.24 | 1458.24 |
> | CIFAR-100 | 2317.04 | 2136.53 | 3775.30 |
> | ImageWoof | 408.82 | 404.07 | 558.21 |
> | ImageNet-10 | 894.62 | 765.51 | 1197.12 |
>
> **Table 2 Convergence Time(s) on Quantization Aware Training**
>
> | Dataset  | AdamW | S-Adam (k=2) | S-Adam (k=8) |
> | --- | --- | --- | --- |
> | CIFAR-100 (2-bit) | 15.89 | 14.33 | 17.41 |
> | TinyImageNet (4-bit) | 281.49 | 258.54 | 293.14 |
> | ImageWoof (4-bit) | 34.96 | 34.74 | 34.38 |
> | ImageNet-10 (2-bit) | 74.21 | 73.21 | 103.29 |
>
> **Q1 no bias correction step**
>
> Bias correction in Adam addresses initialization bias; it is orthogonal to the geometric brake. For simplicity we omitted it, but it can be added exactly as in standard Adam without affecting the convergence analysis. We will note this in the final text.
>
> **Q2 The expected value is included in the definition**
>
> The denominator $\mathbb{E}[D_i^2]+\epsilon$ normalizes the LGI to $[0,1)$, making it scale‑invariant and interpretable as a relative instability measure. Using only variance would not yield a bounded quantity and would conflate absolute gradient magnitude with geometry.
>
> **Q3 Overcome computational gradients**
>
> In non‑smooth settings, the gradient may not exist, but we still use the subgradient returned by automatic differentiation. The LGI probes rely only on function evaluations, not on gradients. Thus the algorithm is applicable even when gradients are undefined, as long as a subgradient oracle (e.g., from backpropagation through a non‑smooth operation) is available. This is standard practice in deep learning.
>
> **Q4 Training time**
>
> S-Adam's improved optimization trajectory means it reaches the target accuracy in fewer epochs, offsetting the per-step cost. Therefore, even on a larger dataset, no additional time cost will be incurred.

---

> > ### Author Rebuttal · Reviewer_yhQ9 · 2026-04-03
> >
> > - Completing the proof of Theorem 5.5: I can image that the techniques that you mentioned could solve the problem however this is not trivial to me and needs to be included in the appendix in more detail than the current explanation. If this is provided, I think the proof is then understandable and correct.
> > -You did not mention how you plan to correct the proof of Lemma A.1. where equation (15) is incorrect. As mentioned before I guess the statement of (15) is too strong and thus not needed for the proof but this needs to be correct as this is the central explanation for the main idea of the algorithm.
> > - I think, including your further explanation, the correction of inconsistencies in Appendix F would clarify all remaining questions in this proof.
> > -Thanks a lot for the detailed explanation regarding training time. I was wondering if I missed in your paper where you are mentioning that $k=2$ is actually a very interesting case. How can you compare it to $k=1$, which you highlighted in table 2?
> >
> > -Q1: Thank you, sounds reasonable
> >
> > -Q2: I can understand the idea to normalize it as well as the mentioned benefit. However, I guess with some extra work or assumptions it could be possible to also bound in a reasonable way as you already showed that it is bounded in Corollary B.1
> >
> > -Q3: Thank you, I guess analyzing or extending the availability of a subgradient oracle is beyond the scope of this paper. So I am fine with it.
> >
> > -Q4: I understand that the algorithm benefits from the improved trajectory but at least for me it is not clear why this should completely annihilate the extra time per step (I also would not expect it to be the case). If I think of a setting where the landscape is simple enough such that Adam already chooses an optimal trajectory, then I expect S-Adam to take significantly longer training time; obviously this is not a case where one would use S-Adam. I think table 1 and table 3 indicate similar results. Maybe you can think of giving an upper bound or an expected behavior for very large training sets.

---

> > > ### Author Response · Authors · 2026-04-06
> > >
> > > Thanks for your follow-up questions. Here we give more detailed answers.
> > >
> > > **W1:**
> > >
> > > S-Adam's update fits the canonical DSA form (eq2&3 of [1]):
> > > $$x_{t+1} = x_t + \eta_t \left[ F(x_t) + M_{t+1} + R_{t+1} \right]$$
> > > where $F(x_t) \in -\bar{\alpha}(x_t)\mathcal{H}(x_t)$ (set-valued drift, satisfying $\mathbb{E}[Y_{n+1}|\mathcal{F}n] \in M(X_n)$),$M{t+1}$ is a zero-mean martingale difference noise, and $R_{t+1}$ is a vanishing remainder.
> > >
> > > We construct a standard affine interpolated trajectory $X(t)$ (matching the asymptotic pseudotrajectory (APT) def in [1]), with time scaling $\tau_n = \sum_{i=1}^n \eta_i$.
> > >
> > > All conditions of the reference are satisfied: Step sizes meet Robbins-Monro conditions;Iterates are almost surely bounded;Noise vanishes asymptotically;The drift map satisfies Hypothesis 2.1 [1].
> > >
> > > By Properties 1-2 of [1], $X(t)$ is an APT of the mean-field flow, and converges almost surely to solutions of the differential inclusion:
> > > $$\frac{dx}{dt} \in -\bar{\alpha}(x) \cdot \mathcal{H}(x)$$
> > >
> > > By the boundedness assumption, the limit set $L(\{x_t\})$ of the S-Adam sequence $\{x_t\}$ is almost surely (a.s.) non-empty, compact and connected.
> > >
> > > Step1 proves the continuous interpolation $X(t)$ of S-Adam iterates is an Asymptotic Pseudotrajectory (APT) of $\frac{dx}{dt} \in -\bar{\alpha}(x)\mathcal{H}(x)$. By **Theorem 5.7** of [2], the limit set of an APT is an internally chain transitive invariant set of the mean-field flow.
> > >
> > > Step2 verifies $f$ is a **strict Lyapunov function** for the differential inclusion.
> > >
> > > By C.1, for any trajectory $x(t)$ and almost all $t$, there exists **some** $\xi \in \partial_C f(x(t))$ such that:
> > >
> > > $$
> > > \frac{d}{dt}f(x(t)) = \langle \xi, \dot{x}(t) \rangle \le -C \cdot \text{dist}\left(0, \partial_C f(x(t))\right)^2
> > > $$ where $C > 0$ is a constant.  Its derivative is zero iff $x$ lies in the Clarke stationary set $\Lambda = \{x \mid 0 \in \partial_C f(x)\}$.
> > >
> > > **Invariance Principle & Final Conclusion**
> > > By Prop.6.4 of [2], any internally chain transitive set of a system with a strict Lyapunov function must be contained in the set where the Lyapunov function is constant. Thus $L(\{x_t\}) \subseteq \Lambda$, yielding:
> > > $$\liminf_{t \to \infty} \text{dist}(0, \partial_C f(x_t)) = 0 \quad \text{a.s.}$$
> > > This completes the proof of Theorem 5.5.
> > >
> > > Eq15 was omitted in the rebuttal; the derivation is as follows:
> > >
> > > We have $f^\circ(x;u) = \max_{g_1, g_2 \in S} \langle g_1 - g_2, u \rangle$. (a definition error in the original text), for any $g_1, g_2 \in S$,
> > > \begin{equation}
> > > \langle g_1 - g_2, u \rangle \le \|g_1 - g_2\|\cdot\|u\| \le \text{diam}(S).
> > > \end{equation}
> > > Therefore, $f^\circ(x;u) \le \text{diam}(S)$. Squaring both sides and taking the expectation over the uniform distribution on the sphere yields:
> > > $$\mathbb{E}_u[f^\circ(x;u)^2] \le \text{diam}(S)^2$$
> > >
> > > Since the square of the expectation is non-negative, the variance satisfies
> > > \begin{eqnarray}
> > > \text{Var}_u\left[f^\circ(x;u)\right] & = & \mathbb{E}_u\left[\left(f^\circ(x;u)\right)^2\right] - \left(\mathbb{E}_u\left[f^\circ(x;u)\right]\right)^2 \nonumber \\
> > > & \le & \text{diam}(S)^2.
> > > \end{eqnarray}
> > >
> > > [1]Benaïm,M.(2005).Stochastic approximations and differential inclusions.*SIAM Journal on Control and Optimization*,328-348.
> > > [2]Benaïm,M.(1999).Dynamics of stochastic approximation algorithms. *Lecture Notes in Mathematics,*1709,1–68.
> > >
> > > **W3:**
> > >
> > > **If k=1, our S-Adam reduces to standard AdamW, as the directional gradient variance vanishes with only one probing direction.  S-Adam (k=2) achieves higher accuracy at similar training cost.**
> > >
> > > **Q2:**
> > >
> > > We appreciate the reviewer’s insightful suggestion on whether the raw variance can be bounded with additional assumptions. However, this approach is not practically viable in real-world optimization. To derive a uniform global upper bound for the raw variance Var({Di}), one must assume globally bounded gradient norms—i.e., a finite constant G such that ||∇f(x)||≤G for all x. Unfortunately, this assumption is untenable in non-convex optimization: neural network loss functions have no universal global upper bound on gradient norms, which typically fluctuate severely during training.
> > >
> > > **Q4:**
> > >
> > > In S-Adam, the extra time per step depends on the number k. It directly dictates the training time because the algorithm must perform $k$ additional forward passes during every single iteration to estimate the instability of the loss landscape.  Therefore, if the function is simple enough, we can take smaller number k, which speeds up the training. For your information, we add one experiment on a large-scale training set Imagenet with 8 Bit QAT on 4 NVIDIA A800 GPUs , and the results are shown below.  We **can see that our S-Adam achieves better accuracy with k=2 but the time is close to AdamW**.
> > > | Optimizer | Conv. Time (s) | Acc (%) |
> > > | --- | --- | --- |
> > > | S-Adam k=8 | 12770 | 11.2 |
> > > | S-Adam k=2 | 8297 | 11.0 |
> > > | AdamW | 8,186 | 8.6 |
> > > | Prox-SGD | 12,322 | 9.7 |
> > > | RMSProp | Diverged | 1.7 |

---

### Official Review · Reviewer_K5iC · 2026-03-13

**Soundness:** 3
**Presentation:** 3
**Significance:** 2
**Originality:** 2
**Overall Recommendation:** 4
**Confidence:** 3

**Summary:**

This paper proposes a new optimization algorithm called Singularity-aware Adam (S-Adam) that targets the problem of gradient chattering, which may arise in non-smooth optimization settings or in training scenarios such as low-bit quantization. The method introduces a metric called Local Geometric Instability (LGI) to measure instability in the local loss landscape through the variance of randomized directional derivatives. This metric is then used to manipulate the effective step size through an adaptive damping mechanism. The authors evaluate the proposed method in extreme optimization settings, including training on designed non-smooth manifolds and quantization-aware training, using models such as ResNet-18 across several datasets.

**Compliance With Llm Reviewing Policy:**

Affirmed.

**Final Justification:**

I appreciate authors' effort and responses during the rebuttal. They have extended the experiment scope and resolve most of my concerns regarding the theoretical aspect of the work and therefore, I decide to raise my score to 4.

**Key Questions For Authors:**

Questions:

1. The main difference between this method and others is that it scale the step size according to the LGI criterion. Can we do the same with just smaller learning rate or larger batch size?

2. The LGI metric is based on the variance of randomized directional derivatives. However, in stochastic training settings the variance may also arise from mini-batch noise rather than non-smooth geometry. How does the method distinguish between stochastic gradient noise and true geometric instability in the loss landscape?

**Limitations:**

yes

**Strengths And Weaknesses:**

Strength:

1. Clear theoretical analysis: The work provide lower and upper bound of the subdifferential Diameter through the variance of the random directional derivatives. This motivate the use Local Geometric Instability which also serve as damping factor in the algorithm proposed. The idea is clear and interesting and offer new perspective to estimate the subdifferential.

2. Good experiment performance: The experimental results show that the proposed optimizer can outperform baseline methods such as AdamW and Prox-SGD in challenging scenarios including quantization-aware training and extremely small batch sizes. These results suggest that the proposed damping mechanism can effectively mitigate gradient oscillation in highly non-smooth training regimes.

Weakness:

1. The tightness of the remain unchecked: The theoretical analysis relies on bounds connecting the variance of directional derivatives with the diameter of the Clarke subdifferential. However, the tightness of this approximation is not empirically validated. In particular, when the mean magnitude of the directional derivatives is large or when the model dimension increases, the bound may become loose. Under such circumstances, the LGI metric may not faithfully represent the true subgradient variability. It would be helpful if the authors could provide empirical evidence demonstrating that the estimated variance closely tracks the true Clarke subdifferential in practice.

2. The setting of the experiments can be too extreme to reflect true learning scenario: Several experiments are conducted under highly extreme conditions, such as very small batch sizes or training on artificially constructed non-smooth manifolds. While these settings are useful for illustrating the behavior of the algorithm, it is unclear whether the observed performance improvements would transfer to more standard training scenarios. For example, the accuracy values reported in Table 1 are significantly lower than typical CIFAR-100 training performance (normally around ~80% accuracy), making it difficult to interpret the practical significance of the results.

3. The scale and diversity of the experiments can be improved: The experimental evaluation could be strengthened by including a wider range of datasets and models. For optimizer studies, large-scale benchmarks such as ImageNet are often used to demonstrate robustness and scalability. In addition, the comparison currently includes only two baseline methods. It would be beneficial to compare against a broader set of optimizers, particularly methods that also aim to stabilize training through variance control, such as RMSProp or other adaptive gradient methods.

4. The training time is very long compared to even normal training: One potential application of the proposed method is low-bit or quantization-aware training. However, the algorithm introduces additional forward passes for directional probing, which significantly increases training time. According to the reported results, the training cost is substantially higher than standard training. This overhead may limit the practical applicability of the method in scenarios where training efficiency is critical.

---

> ### Author Rebuttal · Authors · 2026-03-31
>
> We sincerely thank the reviewer for their thoughtful assessment and constructive feedback.
>
> **W1. Tightness of the variance–subdifferential bound**
>
> We agree that empirical validation of the bound would strengthen the work. However, the theoretical result in Lemma A.1 establishes two-sided bounds with constants $c_1$ and $c_2$. While $c_2$ scales with dimension, in practice the LGI metric is used as a relative indicator rather than an absolute estimator. Crucially, the damping mechanism only requires the relative magnitude of LGI across iterations—not an exact estimate of the subdifferential diameter. In fact, the exponential brake $e^{-\lambda\rho_t}$ is monotonic in $\rho_t$, so as long as $\rho_t$ preserves the ordering of instability, the adaptive behavior remains valid.
>
> We also note that in Appendix F, we derive an alternative interpretation linking LGI to the relative curvature $\kappa_\delta$ of the smoothed surrogate, showing that LGI captures geometric structure even when the subdifferential diameter bound is loose. Empirically, the strong correlation $R^2 = 0.82$ between LGI and performance improvement across diverse settings supports its practical fidelity.
>
> **W2. Extremeness of experimental settings**
>
> Our algo is towards stable non-smooth optimization. To verify this, we run experiments on these extreme cases (i.e., low-bit quantization and small batch size), where gradient jitter is severe. To be noted, these seemingly extreme settings are widely required in practical application (e.g., model lightweighting and deployment on edge devices). For your reference, we also run our algo in normal cases (e.g., batch size = 64, without low-bit quantization ), and find it is also effective (compared to AdamW, our S-Adam consistently increases accuracy across all datasets: from 67.43% to 74.18% on CIFAR100, 90.62% to 94.18% on CIFAR10, and 79.82% to 84.60% on Imagewoof (in Table 4 in the paper).
>
>
> **W3. Scale and diversity of experiments**
>
> We add dataset ImageNet-10 and **baseline** RMSProp (lr=0.001) under Table 1 and Table 2. We can see our proposed S-Adam achieves best accuracy.
>
> _Table 1: Accuracy (%) of_ Quantization Aware Training
>
> |Datasets|Prox-SGD|RMSProp|AdamW|S-Adam (k=2)|S-Adam (k=8)|
> |---|---|---|---|---|---|
> |CIFAR-100 (2-bit)|14.13|11.77|15.94|18.11|18.67|
> |TinyImageNet (4-bit)|19.89|13.95|18.91|22.81|23.18|
> |ImageWoof (4-bit)|31.18|34.33|33.24|35.57|35.19|
> |ImageNet-10 (2-bit)|37.40|41.80|46.60|49.40|50.80|
>
> _Table 2: Accuracy (%)_ (ResNet, Batch Size = 2)
>
> |Datasets|Prox-SGD|RMSProp|AdamW|S-Adam (k=2)|S-Adam (k=8)|
> |---|---|---|---|---|---|
> |CIFAR-10|79.44|31.17|80.48|85.46|86.13|
> |CIFAR-100|48.24|1.00|51.12|55.84|56.11|
> |ImageWoof|50.22|12.01|51.18|74.21|75.87|
> |ImageNet-10|79.20|22.20|78.40|88.80|89.20|
>
> **W4. Training time overhead**
>
> While the experimental results in the paper record the time taken to complete the same number of epochs, this does not reflect the total training time. However, this is not the case. The extra forward passes do increase the _per-iteration_ cost; however, because S-Adam takes a vastly superior optimization trajectory, it requires significantly fewer total epochs to converge(In appendix figure5). When evaluating the total wall-clock time to reach convergence as shown in Table 3, **S-Adam is highly efficient**. At $k=8$,it is comparable to AdamW, while at $k=2$,the total wall-clock time is actually shorter than AdamW across the majority of our benchmarks. Furthermore, we can see that setting the probing directions to k=2 achieves almost identical accuracy to k=8 as shown in abovementioned Table 1 and 2. Notably, the RMSProp and Prox-SGD fails to converge.
>
> Table 3: Convergence Time(s) on QAT
>
> |Dataset|AdamW|S-Adam (k=2)|S-Adam (k=8)|
> |---|---|---|---|
> |CIFAR-100 (2-bit)|15.89|14.33|17.41|
> |TinyImageNet (4-bit)|281.49|258.54|293.14|
> |ImageWoof (4-bit)|34.96|34.74|34.38|
> |ImageNet-10 (2-bit)|74.21|73.21|103.29|
>
> **Q1. Can we achieve the same effect with just smaller learning rate or larger batch size?**
>
> **No.** Smaller learning rates scale updates uniformly, slowing convergence in well-behaved regions. S-Adam is parameter-wise, adaptively damping only those traversing non-smooth geometry. Larger batch sizes merely reduce stochastic noise; they cannot fix geometric non-smoothness, where "chattering" occurs even with full-batch gradients. LGI uniquely identifies this geometric instability, which persists regardless of batch size or noise levels.
>
> **Q2. How does the method distinguish stochastic gradient noise from geometric instability?**
>
> S-Adam distinguishes these by
>
> (1) using a fixed mini-batch for all k directional probes at each step, ensuring captured variance is purely geometric rather than sampling-based; and
>
> (2) exploiting the fact that stochastic noise is isotropic (random), while geometric instability is anisotropic (structured).
>
> LGI targets the latter, identifying the direction-dependent "kinks" that persist regardless of gradient noise.

---

> > ### Author Rebuttal · Reviewer_K5iC · 2026-04-03
> >
> > I thank the authors for their responses. There are still several concerns of mine remained unsolved:
> >
> > 1. Scale of empirical validation.
> > The addition of ImageNet-10 experiments is appreciated. However, this setting is still significantly simplified compared to full ImageNet, which remains a standard benchmark for evaluating optimization methods at scale.
> >
> > 2. Distinction between stochastic noise and geometric instability.
> > The explanation provided is intuitively appealing. However, it is not supported by controlled experiments isolating these two factors. It remains unclear whether the observed improvements are specifically due to capturing geometric non-smoothness, or could be attributed to other effects.  Additionally, the assumption that stochastic gradient noise is isotropic may not hold in practice. Prior work suggests that gradient noise can exhibit structured, anisotropic behavior[1]. This further complicates the interpretation and makes it unclear whether the current experimental design can effectively distinguish between stochastic and geometric effects.
> >
> > [1] Gregory Dexter, Borja Ocejo, Sathiya Keerthi, Aman Gupta, Ayan Acharya, Rajiv Khanna, A precise characterization of sgd stability using loss surface geometry. ICLR 2024

---

> > > ### Author Response · Authors · 2026-04-06
> > >
> > > Q1: Thanks for your follow-up questions. To show the effectiveness of our S-Adam, we add one experiment on the large-scale dataset Imagenet (8bit QAT) , and the results are shown in the following table. We can clearly see that S-Adam achieves substantially higher validation accuracy than mainstream optimizers including AdamW and Prox-SGD in this large-scale training set, fully validating its exceptional performance and robustness in non-smooth optimization landscapes. Furthermore, with a reduced sampling rate (k=2), S-Adam’s computational overhead is fully comparable to that of AdamW, delivering superior accuracy with negligible additional training cost. We can see that **our S-Adam is also effective in the large-scale training**.
> > >
> > > | Optimizer | Conv. Time (s) | Acc (%) |
> > > | --- | --- | --- |
> > > | S-Adam k=8 | 12770 | 11.24 |
> > > | S-Adam k=2 | 8297 | 10.97 |
> > > | AdamW | 8,186 | 8.63 |
> > > | Prox-SGD | 12,322 | 9.66 |
> > > | RMSProp | Diverged | 1.68 |
> > >
> > > Q2：
> > >
> > > We think that the effectiveness of geometric instability is obvious in our S-Aadm. Therefore, we can not understand the reviewer’s question about the controlled experiments isolating stochastic gradient noise and geometric instability.  We feel sorry that we do not explain this clearly in last rebuttal. Here, we add more details.
> > >
> > > (1)  In algorithm 1: The geometric probe uses **deterministic loss evaluations, not stochastic gradients**, so it is immune to mini-batch noise—even anisotropic noise. In details, the learning rate is adjusted adaptively only by the geometric indicator (Line 6, 7 and 14), and Adam’s updates handle gradient noise separately (Line 8 -12). Anisotropic noise affects gradient steps, but not the geometric measurement itself. Thus, **the improvement is indeed caused by capturing geometric non-smoothness**, and our algo clearly distinguishes geometric and stochastic effects. By combining variance estimation across random directions and normalization, it significantly amplifies the geometric signal, converting the anisotropic information embedded in the high-dimensional curvature structure into an easy-to-use scalar indicator.
> > >
> > > (2) As established in *A precise characterization of sgd stability using loss surface geometry* (ICLR 2024), the **stochastic gradient noise induced by SGD updates has negligible impact on AdamW**: the second-moment normalization in AdamW inherently whitens anisotropic gradient noise and greatly reduces its influence on optimization dynamics.
> > > Furthermore, our LGI-based step-size modulation relies on **deterministic explicit forward finite difference estimates** (Line 6 of Algorithm 1) to compute directional derivatives, rather than stochastic mini-batch gradients. Therefore**, our S-Adam is not affected by** **stochastic gradient noise.**

---

### Decision · Program_Chairs · 2026-04-30

**Decision:**

Accept (regular)

**Comment:**

This paper proposes a new optimization method, S-Adam, which adaptively controls step sizes based on Local Geometric Instability (LGI) in order to address gradient instability (so-called gradient chattering) arising in non-smooth loss landscapes. The core idea, indirectly capturing the structure of the Clarke subdifferential via randomized finite differences and using this estimate to damp updates, is both novel and interesting. These strengths were recognized by the reviewers. In particular, through the rebuttal process, the authors expanded the experimental scope (e.g., additional datasets and baselines) and clarified theoretical aspects, which helped alleviate several concerns raised by the reviewers. As a result, a majority of the reviewers ultimately leaned toward acceptance.

At the same time, several important concerns remain. First, it is not yet fully clear whether the observed performance gains stem from detecting non-smooth geometric structure, or can instead be attributed to an effective adjustment of the learning rate. Second, the theoretical section contains omissions and inconsistencies in the proofs; in particular, certain steps of the main theorem and the rigor of some lemmas require completion and correction in the final version. Third, the experimental evaluation is primarily focused on regimes emphasizing non-smoothness, and further validation is needed to establish effectiveness and scalability in more standard large-scale settings.

Taking these points into account, I recommend acceptance of this paper, conditional on the following improvements: completion and clarification of the proofs of the main theorem and supporting lemmas; a more rigorous and clearly presented analysis of the role of LGI (distinguishing geometric effects from learning-rate effects); and more careful experimental validation, including clearer baseline comparisons. With these improvements, the impact of this work is expected to increase further.